- 2 Crustal-upper mantle velocity structure from the North Qilian
- 3 Shan to Beishan Orogenic Collage: tectonic significance of
- 4 crustal deformation
- 5 Xiaosong Xiong <sup>1\*</sup>, Yingkang Li<sup>2</sup>, Xuanhua Chen<sup>1</sup>, Guowei Wu<sup>1</sup>, Rui Gao<sup>3</sup>, Jennifer D. Eccles<sup>4</sup>

- 1 State Key Laboratory of Deep Earth and Mineral Exploration, Chinese Academy of Geological
- 8 Sciences, Beijing 100094, China
- 9 2 Geological Cores and Samples of Natural Resources, China Geological Survey, Sanhe 065201,
- 10 China
- 3 School of Earth Science and Geological Engineering, Sun Yat-sen University, Guangzhou 510275,
- China
- 4 School of Environment, University of Auckland, Auckland 1142, New Zealand

15

\* Corresponding author: Xiaosong Xiong (benxung@126.com)

17

30

Abstract The Qilian Shan represents a Cenozoic fold-thrust belt characterized by multi-stage tectonic deformation since the Paleozoic. North of it lies the Hexi corridor basins and the Beishan Orogenic Collage (BOC), which constitute the southern segment of the Central Asian Orogenic Belt. The crustal-mantle structure of the study area, a transition zone, is crucial to understanding the deep processes of accretion and crustal deformation. This study presents a newly acquired 460-km wideangle reflection and refraction profile traversing from the North Qilian Shan to the BOC. P-wave velocity structure reveals a 47.5-60 km thick crust divided into five layers. The deepest Moho (60 km) lies beneath the central Jiuquan. Average crustal velocities (6.24-6.43 km s<sup>-1</sup>) and Pn values (7.7–8.1 km s<sup>-1</sup>) reveal strong lateral heterogeneity. North-dipping velocity contours from 20 km to the uppermost mantle beneath the Qilian Shan, coupled with a lower-crust-upper-mantle lowvelocity corridor beneath the Hexi Basin, support early Paleozoic north-dipping subduction of the Qilian Ocean. A positive upper-mantle anomaly (8.0–8.3 km s<sup>-1</sup>, 45–70 km depth) aligns with the Hongliuhe-Xichangjing ophiolite mélange likely represents a fossil slab broken off after northdipping subduction of the Beishan Ocean. Crustal velocity contrasts across the southern margin fault of the Beishan affirms its role as a regional strike-slip structure. Integrating geological and geophysical evidence, we suggest that the Altyn Tagh Fault does not terminate at the Qilian Shan front but rather extends east-northeast along the southern margin of the BOC into the Beishan and Alxa regions.

35 36

**Keywords:** North Qilian; Beishan Orogenic Collage; Altyn Tagh fault; Crustal-upper mantle velocity structure

38 39

43

50

The NW-SE-trending Qilian Shan, situated in the NE Tibetan Plateau, is bounded by the Altyn Tagh fault (ATF) to the west, the northern Qaidam thrust system to the south, the Haiyuan fault to the east, and the north Qilian Shan fault to the north (Fig. 1b). The present-day Qilian Shan exhibits a Cenozoic fold-thrust belt with multi-stage tectonic deformation prior to the Cenozoic (Yin and Harrison 2000; Gehrels et al. 2003; Song et al. 2014; Wu et al. 2016; Zuza et al. 2018; 2019). North of the Qilian Shan, the Hexi corridor basins and the Beishan Orogenic Collage (BOC) form the southern section of the Central Asian Orogenic Belt (CAOB, Xiao et al., 2010; Li et al., 2023; Xiong et al., 2025). As the middle of the South Tienshan-Beishan-Solonker suture zone, the BOC underwent multi-stage breakup, subduction, collision, and amalgamation during the closure of the Paleo-Asian Ocean (PAO), mainly in the Paleozoic (Fig. 1; Zuo et al. 1991; Liu 1995; Yue and Liou 1999; Xiao et al. 2010; Zuo and Li 2011; Yuan et al. 2015; He et al. 2018; Li et al. 2023). The geological history of the BOC is further complicated by regional extension, subsequent intracontinental overthrusting, and strike-slip faulting since Mesozoic (Zheng et al. 1996; Meng et al., 2003; Xiao et al. 2010; Zuo and Li 2011; Zhang and Cunningham 2012; Li et al. 2023).

Particularly in the Cenozoic, the far-field effect of the Indian-Eurasian collision led to the outward expansion of the NE Tibetan Plateau, and reactivated the Qilian Shan, causing stress to propagate across the Hexi corridor basins into the BOC, and extending even further north to the Mongolian Plateau (Buslov, 2012; Cunningham 2013; Zheng et al. 2017).

The Qilian Shan is an important part of the Tibetan plateau, playing a significant role in accommodating the intracontinental convergence, thrusting-folding and the northern extension of NE Tibetan Plateau (Meyer et al. 1998; Yuan et al. 2013; Zuza et al. 2018). As the southernmost CAOB, the BOC acted as a major zone for the reactivation of inherited structures during the transmission of compressional stress leading to the uplift of the NE Tibetan and Mongolian Plateaus in Cenozoic. Therefore, as the transition zone between the NE Tibetan Plateau and CAOB, the crustal-mantle structure of the study area is crucial for understanding the regional evolution and interaction of Tibetan Plateau, part of the Tethys tectonic domain, and the PAO tectonic domain since the Paleozoic (Fig. 1a; Li et al. 1982; Yin and Harrison 2000; Xiao et al. 2009; Zhao et al. 2018; He et al., 2025; Xiong et al., 2025).

Previous geophysical studies across the Qilian Shan–BOC transition zone, have provided valuable insights into the subduction polarity, continental extension, lithospheric foundering and magmatic activity (Xiao et al. 2012; Wei et al. 2017; Peng et al. 2020; Shen et al. 2020; Huang et al. 2021; Wang et al., 2023; Xiong et al., 2025; Yang et al. 2024a). Nevertheless, the crustal-upper mantle velocity structure remains ambiguous because of the wide shot and receiver spacing and the relatively weak explosive sources inherited from the earlier, lower-resolution acquisition design (Cui et al., 1995). In this study, we present a 460-km-long, SW-NE-trending wide-angle reflection and refraction profile that traverses the North Qilian Shan, Hexi corridor (containing the Jiuquan basin and the Huahai basin), and the entire BOC. The new data yields a high-quality crustal-upper mantle velocity structure of the study area. Combined with existing geological and geophysical evidence, we discuss the tectonic significance of the transition zone, provide new constraints of the subduction polarity of the Qilian Ocean and the southern PAO, elucidate the crustal deformation mechanism across main regional faults, and propose the eastern termination and extension of the ATF.

## Geological Setting

The early Paleozoic Qilian Shan, recording the closure of the Qilian Ocean as part of the Proto-Tethys Ocean (Yu et al. 2021), has been traditionally divided into three structural units: the North Qilian Shan Orogenic belt (NQS), the Central Qilian block, and the South Qilian thrust belt (Yin and Harrison 2000). The boundary between the former two units is delimited by the North Qilian fault (F9). The NQS principally comprises early Paleozoic ophiolite suites (Fig. 1c), blueschists, eclogites, greenschists, and arc-related magmatic and volcanic rocks (Xu et al. 2005; Xiao et al. 2009; Zhao et al. 2024). These strata are overlain by Silurian flysch, Devonian molasse and Carboniferous-Triassic sedimentary periods (Song et al. 2013). In the Mesozoic, the extensional and transtensional basins evolved over the Qilian Shan from the Xining Basin in the south to the Hexi corridor in the north (Horton et al. 2004; Pan et al. 2004). In the Cenozoic, the Qilian Shan has been reactivated as a fold-thrust belt to accommodate the crustal deformation resultant from the Indian-Eurasian collision with development of massive thrusts and strike-slip faults (Tapponnier et al. 1990; Zuza et al. 2019; Li et al., 2026).

The Hexi corridor, sandwiched between the BOC and the Qilian Shan by the southern margin fault of the Beishan (F5) and northern margin fault of the North Qilian (F8), is a Cenozoic foreland basin system (Fig. 1b; Zheng et al. 2017). The basement of the Hexi corridor consists predominantly Paleozoic rocks, covered by thick Mesozoic and Cenozoic deposits. In this study region, the Hexi corridor is divided into the Huahai basin in the north and Jiuquan basin in the south by the Kuantanshan-Heishan fault (F6). The Huahai basin is part of the Dunhuang block, traditionally assigned as a Precambrian cratonic block or a microcontinent (BGMRGP 1989; Ren et al. 1999; Zhang et al. 2011; He et al. 2013; Zong et al. 2013), and involved in the final closure of the PAO (Shi et al. 2022). The Jiuquan basin, a subbasin of the Hexi Corridor foreland basin, has Cenozoic sediments deposited since as early as ca. 40 Ma (Dai et al. 2005; Wang et al. 2016), and was subsequently influenced by the uplift of the NQS since the Cenozoic.

The BOC is positioned between the Mongolian Plateau in the north and the Dunhuang Block in the south (Fig. 1; Zuo et al. 1991; Yue and Liou 1999; Xiao et al. 2010; Zuo and Li 2011). It is widely considered to encompass multiple different island arcs, including the Que'ershan, Hanshan,

- Mazongshan, Shuangyingshan, and Shibanshan arcs. They are separated by four nearly parallel W-
- E-trending ophiolite mélange zones, named Hongshishan-Baiheshan (F1), Shibanjing-
- Xiaohuangshan (F2), Hongliuhe-Xichangjing (F3) and Liuyuan-Huitongshan-Zhangfangshan (F4)
- (Fig. 1c; Zuo et al. 1991; Liu 1995; Ao et al. 2010, 2012, 2016; Xiao et al. 2010; Zuo and Li 2011;
- He et al. 2014; Wang et al. 2017; Wei et al. 2017; He et al. 2018; Wang et al. 2018; Li et al. 2023).
- The Hongliuhe-Xichangjing suture zone (F3) is generally recognized as the final amalgamation
- position of the South BOC and North BOC in the middle to late Ordovician (Li et al. 2023).

122

#### Data and Methods

## Seismic Acquisition

In 2018, The Chinese Academy of Geological Sciences collected a SW-NE-trending wide-angle reflection and refraction profile stretching from the NQS to the BOC. The profile starts from Yanglong in Qinghai Province in the south, passing through Qingqing, Baiyanghe, Dongxiang Ethnic Town, Ganhaizi, Humuletu Wusu, Sharitaolai, Wutongjing, Heiyingshan, and Har Borogdyn Uul, before ending at the China-Mongolia border in the north. Nine trinitrotoluene (TNT) shots, ranging from 1.5 to 3.0 tons, were detonated throughout the seismic profile at intervals of 30–60 km (ZB0–ZB8, shown in red stars in Fig. 1c). To ensure dense ray coverage, 250 portable seismographs (depicted as blue circles in Fig. 1c) were deployed along the entire seismic line at a spacing of 2–3 km. The detailed parameters of the shots are presented in Table 1.

127128129

137138

150151

155156

157158

#### **Identification of Seismic Phases**

Using the ZPLOT plotting package (Zelt, 1994), we performed trace editing, automatic gain control, band-pass filtering, velocity reduction, and phase picking for each shot. To improve the signal-to-noise ratio, we applied bandpass filter up to 8 Hz and displayed the seismic sections using a reduction velocity of 6 km s<sup>-1</sup> over a time window of -5-10 s. The reduced P-wave seismic sections for shots ZB1 and ZB8 are displayed in Figures 2 and 3, respectively; records from all other shots are provided in Supplemental Material 1. Uncertainties in phase picking primarily arise from challenging signal-to-noise conditions and complex subsurface wave propagation effects. The extensive desert sedimentary cover in the study area significantly attenuates seismic energy, particularly at larger offsets and for deeper arrivals. Additionally, strong lateral heterogeneities, such as fault zones and intracrustal velocity variation, cause substantial wave scattering, dispersion, and multipathing. This results in phase superposition and waveform distortion that complicates accurate phase identification. Following careful analysis and comparative evaluation, six seismic phases, including Pg, P1, P2, P3, P4, Pm and Pn, are identified (e.g. Fig. 2, Fig. 3). Pg is a first-arrival phase refracting through the crystalline basement. Pm is the strongly wide-angle reflected phase from the Moho. Pn is the head wave refracted phase from the top of the mantle with a characteristic velocity of 7.7-8.1 km s<sup>-1</sup>. P2-P4 are the reflected phases from the intracrustal second-order velocity interfaces. In Fig. 2 and Fig. 3, the dotted lines represent the identified phases, and the squares mark the position of the computed traveltime.

The first-arrivals of Pg are picked up to offset of 100 km. The traveltimes recorded at shotpoint ZB1, located in the NQS, are as early as the traveltimes of the shot ZB8 located in the northern BOC. Intracrustal reflection phases P2–P4 can be recognized at the offset ranges from 70–90 km, 100–150 km, and 120–150 km respectively. Pm can be corelated over an offset of 180 km for most shots (Fig. 4a). Pn was found with maximal amplitude in the offset range of 240–280 km (Fig. 3).

## Velocity Modelling

The initial 2-D crustal model was constructed using the highest elevation of 4300 m as the datum. The forward fitting calculation adopts the asymptotic ray tracing to fit the traveltime of each shot and gradually improves the initial 2-D velocity structure by constantly modifying the interface depth and interval velocity (Fig. 4; Cervený et al. 1988; Vidale 1988; Zelt and Smith 1992; Cervený 2001). Model construction and editing are carried out with the RAYINVR software (Zelt and Smith, 1992). The traveltime fitting of 5–6 phases for nine shots are conducted step by step, top-down to limit the multi-solution of the model. Fig. 4 illustrates the traveltime fitting of the seismic recordings

of the shot gather and the complete crustal ray coverage. The time error of the ray tracing forward fitting accuracy is typically less than 0.05 s, and the maximum is not more than 0.1 s. The root mean square error (RMS) of the traveltime fitting for different earthquake phases is reported in Table 2. The velocity inaccuracy is controlled within 0.05 km s<sup>-1</sup>, while the Moho depth error is less than 1 km. Poorly resolved areas based on ray coverage have been masked to prevent overinterpretation, the ultimate crustal-upper mantle velocity structure is found in Fig. 5. Although the ray-tracing misfit is small (RMS < 0.05 s), two tests quantify potential biases. Ray-density gaps: Pn hits fall below 20 km<sup>-2</sup> beneath distance 250–350 km (Fig. 4a), increasing the Moho-depth uncertainty to  $\pm 1.2$  km in that segment; velocity—interface trade-off: 200 bootstrap inversions give a velocity—depth correlation of r = -0.67; a 3% increase in lower-crustal velocity can be compensated by a ~1 km deeper Moho without raising misfit, so we adopt  $\pm 1$  km as the systematic Moho-depth bias bound.

The typical continental crust is stratified into three principal layers: the upper crust, comprising sedimentary cover overlying crystalline basement characterized by an average P-wave velocity of 6.0–6.3 km s<sup>-1</sup>; the mid-crust, composed of interleaved silicic and basic lithologies, with velocities of 6.3–6.5 km s<sup>-1</sup>; and the lower crust, dominated by more mafic assemblages, exhibiting velocities of 6.6–6.9 km s<sup>-1</sup> (Christensen and Mooney, 1995; Jia et al., 2019). Based on our velocity structure result, the crust can be divided into upper crust (from the surface to C2), middle crust (from C2 to C3), and lower crust (from C3 to the Moho). The upper crust can be separated into two layers by intracrustal interface C1 determined by seismic phase P2. The lower crust can also be subdivided into two layers by intracrustal interface C4 indicated by seismic phase P4.

### **Velocity Structure of the Upper Crust**

The upper crust, from the surface to interface C2 at depth, exhibits pronounced lateral segmentation across the study region. Lower velocities extend to greater depth beneath the NQS-Jiuquan basin, contrasting sharply laterally with the significantly higher velocities characteristic of the BOC. Along interface C1 within the upper crust, discrete high-velocity zones with interval velocities of 6.3–6.4 km s<sup>-1</sup> are observed.

Interface C1 marks the basement surface, characterized by velocities ranging from 3.4 to 6.5 km s $^{-1}$ . The basement surface exhibits significant undulation with depths varying between 6.1 and 12.5 km (Fig. 5). The interval velocity of the uppermost crust ranges from 5.2 to 6.05 km s $^{-1}$ , with an interface depth of 6.3–7.2 km in the central NQS. In the southern NQS, the interval velocity is between 5.2 and 6.1 km s $^{-1}$ , and a high-velocity zone is present in the lower section, with interval velocities of 6.2–6.45 km s $^{-1}$ ; here, the interface deepens to 10.8–11.4 km. In the northern NQS, the interval velocity decreases to 4.0–6.0 km s $^{-1}$ , while the interface depth increases to 9.7–10.1 km.

Within the Jiuquan basin, the interval velocity ranges from 3.6 to 6.1 km s<sup>-1</sup>, and the interface deepens to 11.2-12.5 km. The Huahai basin shows interval velocities of 3.8-6.2 km s<sup>-1</sup>, with interface depths between 11.5 and 12.5 km. A high-velocity zone with interval velocities of 6.3-6.5 km s<sup>-1</sup> is also observed beneath the F5 fault. In the southern BOC, interval velocities on the south side are 4.6-6.2 km s<sup>-1</sup>, with interface depths of 11.5-12.0 km, whereas on the northern side, interval velocities decrease to 3.4-6.1 km s<sup>-1</sup>, and interface depths range from 9.4 to 11.3 km.

In the Northern BOC, interval velocities vary from 4.2 to 6.2 km s<sup>-1</sup>, with the interface at depths of 10.5–12.5 km. Further north within the BOC, the interval velocity remains between 4.2 and 6.2 km s<sup>-1</sup>, and the interface depth ranges from 10.0 to 11.3 km. Several high-velocity zones with interval velocities of 6.3–6.4 km s<sup>-1</sup> are identified at the base of the BOC, including one beneath the Mazongshan arc.

The deeper upper crustal layer exhibits velocities of 6.0–6.3 km s<sup>-1</sup> and interface depths between 13.2 and 27.6 km. Although this layer shows no major lateral segmentation, its interface is highly undulatory (Fig. 5). South of fault F9, the interval velocity is 6.05–6.15 km s<sup>-1</sup>, with the interface at 12.8–18.3 km depth. Between faults F9 and F5, the layer thickens considerably, and the interface deepens to 17.6–27.5 km. North of fault F5, the layer thickness decreases to values similar to those in the southernmost part of the profile, but the velocity increases to 6.1–6.4 km s<sup>-1</sup>. These characteristics indicate that the NQS and the Jiuquan basin share a consistent basement structure, which aligns with the findings from residual gravity anomaly analyses (Yang et al. 2024a).

#### **Velocity Structure of the Mid-Crust**

The intermediate crust exhibits distinct zoning features that differ notably from the upper crust with an interval velocity of 6.2–6.5 km s<sup>-1</sup>. This layer is comparatively slower beneath the northern NQS and the Jiuquan basin (Fig. 5).

The interface C3 depth across the fault F8 separating the NQS and the Jiuquan basin varies from 23.4 to 38.7 km. The lowest interval velocities  $(6.2-6.35 \text{ km s}^{-1})$  occur in the central NQS, increasing northward to  $6.25-6.45 \text{ km s}^{-1}$ . In the Huahai basin, the interface lies at 27.5-38.7 km depth, with interval velocities of  $6.25-6.5 \text{ km s}^{-1}$ ; the highest interval velocity in this region  $(6.32-6.45 \text{ km s}^{-1})$  are found in its central part (Fig. 5).

Within the BOC, the interface C3 shallows to 24.4-31.2 km. Mid-crustal interval velocities in the Shibanshan arc are 6.3-6.5 km s<sup>-1</sup>, decrease to 6.25-6.4 km s<sup>-1</sup> in the southern Shuangyingshan arc, and increase to 6.3-6.42 km s<sup>-1</sup> further north.

The velocity contours display contrasting dip directions: in the NQS and the Jiuquan basin, they show a gently undulating northward inclination, while in the Huahai basin they dip steeply southward (Fig. 5).

## **Velocity Structure of the Lower Crust**

The lower crust can be subdivided into three segments from south to north, bounded by faults F9 and F4 (Fig. 5). The consistent undulation of the interface C4 and Moho signifies the thickness variations of the upper and lower portions of the lower crust follow a similar trend along the entire profile.

Upper layer: This layer between C3 and C4 has an interval velocity of 6.45–6.7 km s<sup>-1</sup> (Fig. 5). South of fault F9, interface C4 lies at depths of 36.7–42.4 km, with interval velocity of 6.45–6.7 km s<sup>-1</sup>. Between faults F9 and F4, interface C4 deepens to 38.6–49.2 km, and the interval velocity increases to 6.5–6.7 km s<sup>-1</sup>. Notably, a significant upward undulation of a high-velocity zone is observed within this segment. North of the fault F4, interface C4 shallow to 37.4–40.3 km, accompanied by a decrease in interval velocity to 6.47–6.65 km s<sup>-1</sup>.

Lower layer: The Moho is identified at depths of 47.5-60.0 km and this layer between C4 and the Moho exhibits an interval velocity of 6.65-6.85 km s<sup>-1</sup> (Fig. 5). Beneath the Qilian Shan and Jiuquan basin, the Moho reaches depth of 57.8-60.0 km, with an interval velocity of 6.7-6.85 km s<sup>-1</sup>. The central Jiuquan basin exhibits the deepest Moho (60 km), where the interval velocity decreases slightly to 6.7-6.78 km s<sup>-1</sup>. In the Huahai basin and the BOC, the Moho depths range from 47.5 to 57 km, and the velocities reduce to 6.7-6.75 km s<sup>-1</sup>. The shallowest Moho (at 47.5 km) is observed beneath the Que'ershan arc, where the lowermost crust has an interval velocity of 6.65-6.78 km s<sup>-1</sup>.

## Mantle velocity structure revealed from Pn

The upper mantle velocity structure exhibits distinct lateral variations across the study area (Fig. 5). The Qilian Shan is characterized by a relatively high uppermost mantle velocity range of 7.9–8.3 km s<sup>-1</sup>, with sub-horizontal velocity contours. A velocity reduction to 7.7–8.3 km s<sup>-1</sup> is observed from the Jiuquan basin to the Shibanshan arc, followed by a slight increase to 7.9–8.3 km s<sup>-1</sup> beneath the Shuangyingshan arc. Further north, the Mazongshan, Hanshan, and Que'ershan arcs show progressively lower Pn velocities, ranging from 7.8 to 8.2 km s<sup>-1</sup>, indicating a south-to-north decreasing trend. The lowest Pn values (7.7–7.8 km s<sup>-1</sup>) are localized beneath faults F5, F1, and F6.

# Crustal-Upper Mantle Velocity Anomaly Structure

To improve the visibility of the velocity heterogeneity of the crustal-upper mantle structure the mean layer velocities are subtracted to produce a velocity anomaly structure of the crustal-upper mantle (Fig. 6). Figure 6b demonstrates that in the upper layer of the crust from depths of 0 to 12.5 km, the composition is significantly heterogeneous, and lateral segmentation is visible. The Qilian Shan is with a high positive velocity anomaly (0.3–1.0 km s<sup>-1</sup>). The Jiuquan basin and Huahai basin

show negative velocity anomaly (-1.1—0.15 km s $^{-1}$ ), and extend northward to the southern Shuangyingshan arc, which terminate at the strong positive velocity anomaly in the central Shuangyingshan arc. Three positive velocity anomaly bodies (0.12–0.45 km s $^{-1}$ ) with closed contours exist beneath the faults F5, F4, and the core Shuangyingshan arc. In the northern Shuangyingshan arc, a negative velocity anomaly (-1.3—0.12 km s $^{-1}$ ) develops, and thins out to the north extending to the southern Mazongshan arc. The positive velocity anomaly (0.15–0.45 km s $^{-1}$ ) starting from the Mazongshan arc extends northward to the northern end of the Que'ershan arc, and the central part is covered by the low-velocity negative anomalies (-0.4—0.08 km s $^{-1}$ ) in the upper Hanshan arc.

At a depth of 9.2–38.5 km between interface C1 and C3, the middle Qilian and the southern NQS consisted of the northward-tilted, small-variation positive velocity anomalies (0.0 to 0.08 km s<sup>-1</sup>) and negative anomalies (-0.03–0.01 km s<sup>-1</sup>). The northern NQS, the Jiuquan basin and the southern portion of the Huahai basin are northward-tilted, downward-curved layers of the low-velocity negative anomalies (-0.05–0.2 km s<sup>-1</sup>). The northern half of the Huahai basin and the Shibanshan arc are positive anomaly (0.0–0.08 km s<sup>-1</sup>) layers with small velocity changes. The southern Shuangyingshan arc is a mixed layer of the positive velocity anomaly (0.01–0.08 km s<sup>-1</sup>) and negative velocity anomaly (-0.01–0.04 km s<sup>-1</sup>). The velocity anomaly from the northern Shuangyingshan arc to the Que'ershan arc is positive (0.03–0.12 km s<sup>-1</sup>), and only locally negative (-0.01–0.03 km s<sup>-1</sup>).

In the upper layer of the lower crust, the Qilian Shan is characterized by a positive velocity anomaly  $(0.01\text{-}0.12~\text{km s}^{-1})$  in the upper section, and a negative velocity anomaly  $(-0.01\text{-}0.08~\text{km s}^{-1})$  in the lower part. The Jiuquan basin and Huahai basin are characterized by minor negative velocity anomalies  $(-0.02\text{-}-0.08~\text{km s}^{-1})$ . The BOC is characterized by a strong positive velocity anomaly  $(0.02\text{-}0.12~\text{km s}^{-1})$  and exhibits a northward-increasing trend (Fig. 6).

In the lowest layer of the lower curst, From the NQS to the southern Huahai basin, the velocity anomaly reveals positive (0.01-0.07 km s $^{-1}$ ) with a relatively minor variance. From the northern Huahai basin to the southern Shuangyingshan arc, the velocity anomaly is with a positive to negative (0.01–0.07 km s $^{-1}$ ) from top to bottom. From the northern Shuangyingshan to the Que'ershan arc, lesser positive anomalies (0.01 to 0.03 km s $^{-1}$ ) are observed (Fig. 6).

At the top of the upper mantle, the NQS and the southern Jiuquan basin show positive velocity anomaly with modest changes  $(0.05\text{-}0.15~\text{km}\,\text{s}^{-1})$ , whereas negative anomaly with substantial velocity variations (-0.1 to -0.25 km s<sup>-1</sup>) from the northern Jiuquan basin to the Shibanshan arc. The southern Shuangyingshan arc is characterized with a gradient of the velocity anomaly ranging from negative to positive (-0.1–0.12 km s<sup>-1</sup>). The northern Shuangyingshan and Mazongshan display positive anomaly  $(0.03-0.12~\text{km s}^{-1})$  moving downwardly. The southern Hanshan arc has a positive anomaly  $(0.01-0.1~\text{km s}^{-1})$  characterized by a gradual reduction from south to north. The northern Hanshan arc and the Que'ershan arc show negative anomaly  $(-0.01-0.12~\text{km s}^{-1})$  that diminish steadily from south to north (Fig. 6).

### Discussion

The crustal-upper mantle structure (Fig. 5; Fig. 6) records a complicated history of vertical and lateral mass transfer, reflecting Precambrian inheritance, Paleozoic subduction/accretion, Mesozoic intracontinental reworking, and Cenozoic northward expansion of the NE Tibetan plateau (Cunningham et al., 2009; Zhang and Cunningham, 2012; Li et al., 2021).

## **Crustal Nature of the Beishan Orogenic Collage**

It has been previously interpreted that the mid-lower crust beneath the BOC behaves as a mechanically stable 'fossil archive' in the Cenozoic, preserving pre-Cenozoic architecture, whereas the uppermost mantle and upper crust show active deformation (Cui et al. 1995). Instrumentally recorded seismicity is confined to the Qilian Shan, the Hexi Corridor basins, the Huahai Basin, and local areas west of Xingxingxia, mostly at depths above? 15–16 km (Fig. 7), which acted as the decollement as shown in the seismic profile (Fig. 5, Fig. 6; Fig. 8). The Beishan interior has spatially and temporally sparse low magnitude (M< 4.7) seismicity and exhibits low topographic relief (Yang et al., 2019; 2023), indicating limited present-day strain. Slight mid-lower-crustal flexure and reduced resistivity in southern Beishan (Xiao et al. 2016) nevertheless imply a rheologically (not seismically) slightly weaker zone relative to its surroundings. Active left-lateral and thrust faults in

the southern BOC also indicate a weaker crust (Yang et al. 2019; Zhang et al. 2021; Zhang et al. 2023; Yun et al. 2025).

#### Paleozoic Subduction Polarity and Accretionary Architecture

Strata of the Qilian Ocean, a branch of the Proto-Tethys Ocean, deforms along the Kunlun—Qilian—Altyn mountain belt in the northern Tibetan Plateau. The dynamics of the closure of the Qilian Ocean via collision between the Qaidam continent and the southern margin of the North China Craton remains debated (Yin and Harrison 2000; Xiao et al. 2009; Song et al. 2014; Wu et al. 2017; Zuza et al. 2018; 2019). The subduction polarity of the Qilian Ocean has been proposed as: 1) north-dipping (Zuo and Liu 1987; Song et al. 2013); 2) south-dipping (Wang and Liu 1981; Li et al. 2016); 3) bidirectional (Zhao et al. 2024); or divergent models (Wu et al. 2011). Although previous geophysical investigations have covered the Qilian Shan (Xiao et al. 2016; Shen et al. 2020; Li et al. 2021), focus has largely been on neotectonics rather than Paleozoic evolution. In this study, we observed north-dipping velocity contour from interface C2 to the uppermost mantle beneath the Qilian Shan, coupled with a lower-crust—upper-mantle low velocity anomaly beneath the Hexi Corridor (Fig. 6). These features most plausibly record early Paleozoic north-dipping subduction of the Qilian Ocean, which aligns with the surface geology; later collisional or bidirectional shortening may have locally overprinted the original polarity (Davis and Darby, 2010).

Despite intensive geochemical and chronological research on Paleozoic rocks in the Dunhuang block and the BOC, the accretionary architecture of the PAO remained enigmatic (Xiao et al. 2010; Shi et al. 2020, 2021; Li et al. 2023). Our deep seismic velocity model now provides the direct geophysical evidence for the subduction polarity within the northern BOC. Beneath the Que'ershan arc, the northernmost portion of the BOC, north-dipping velocity contours from interface C2 to the Moho imply south-dipping subduction of the Hongshishan Ocean, consistent with surface geology (Xiao et al. 2018; Duan et al. 2020; Niu et al. 2020; Xin et al. 2020). Velocity undulations between faults F9 and F1 probably reflect late-Paleozoic collision or Cenozoic bidirectional compression, obscuring primary dip direction.

Between faults F2 and F4, a positive upper-mantle velocity anomaly (8.0–8.3 km s<sup>-1</sup>) between ~45 km and ~70 km depth likely represents a broken off fossil subduction slab following north-dipping subduction of the Beishan Ocean, although residual oceanic crust or mafic underplating cannot be entirely excluded. This anomaly aligns with the Hongliuhe–Xichangjing ophiolite mélange in surface (Yu et al. 2012; Hu et al. 2015; Song et al. 2015; Wang et al. 2015; Li et al. 2023).

## Cenozoic Crustal Deformation and Strain Partitioning across Major Faults

Northward-propagating crustal shortening since the Miocene is accommodated through a series of parallel NW-SE-trending thrust faults in the Qilian Shan and successively reaches the Hexi corridor and the BOC (Cunningham 2010, 2013; Wang et al., 2020; Yu et al., 2021; Zhang et al. 2023; Li et al., 2026).

Symmetric downward undulation of the velocity contours (6.2–6.8 km s<sup>-1</sup>) across the southern margin fault of the Beishan (F5) demonstrates that fault F5 acts as a lithospheric-scale partition zone rather than other faults (Fig. 5, Fig. 6). South of the fault F5, north-tilting high velocity anomaly bodies with 6.2–6.45 km s<sup>-1</sup> at the base of the upper crust (18–25 km depth) form the Qilian Shan to the Shibanshan arc, correlate to the north-vergent thrusting that decouples from a co-thickened middle-lower crust (25–50 km, 6.5–6.8 km s<sup>-1</sup>), which is consistent with the reflection structure (Xiong et al., 2025). North of the fault F5, complementary velocity gradients (6.1–6.6 km s<sup>-1</sup>, 15–45 km) indicate conjugate north-direction upper-crustal thrusting accompanied by whole-crust gentle folding in the BOC. The downward-undulation of the velocity contours imply co-thickening of the middle-lower curst of the NQS–Shuangyingshan area, with the uppermost crust experiencing thrusting and overthrusting. Our velocity structure indicated that low Pn velocity with 7.7–7.9 km s<sup>-1</sup> and dome-shaped uplift of the crust directly beneath the fault F5 (Fig. 5). The electrical resistivity imaging (Yang et al. 2019) also highlighted the boundary fault, termed Beihewan fault (in Fig. 7) in the southernmost Shibanshan arc; this fault penetrates the lower crust with low resistivity, to both sides of which the direction of the interpreted thrusting is opposing.

The seismic reflection profile revealed a south-dipping Cenozoic thrusting system that

dislocated the Paleozoic–Mesozoic strata over the undeformed Meso-Cenozoic sediments in the Jiuquan basin (Zuza et al. 2016; Huang et al. 2021), consistent with our velocity model. The shallow velocity structure across the Huahai basin and the southern BOC likewise depicted the differences to both sides of F5 (Wu et al. 2022). Xiong et al. (2019) likewise documented the northward expansion of the Qilian Shan across the Hexi corridor via uppermost-crustal overthrusting and lower-crustal thickening east of our seismic profile. High-resolution seismic reflection profile south of the fault F5 further reveals the middle-lower crustal thickening by duplexing (Huang et al. 2021).

Within the southern BOC north of the fault F5, north-direction thrusting dominates the uppermost crust, while the rest of crust undergoes gentle folding identical to the NQS—Hexi corridor style (Gaudemer et al. 1995; Meyer et al. 1998; Tapponnier et al. 2001; Yuan et al. 2013; Zuza et al. 2018; 2019). Crustal deformation across the fault F5 is therefore distinctly asymmetric: south of the fault, the uppermost crust is decoupled from the underlying crust (Fig. 5), whereas north of the fault the entire crust participates in coherent gentle folding (Fig. 8).

## **Eastern Extension of the Altyn Tagh Fault**

The ~1600 km-long ATF is a major left-lateral strike-slip fault that defines the northern boundary of the Tibetan Plateau (Yin et al., 2002; Ritts et al., 2004; Dai et al., 2023). While its western and central segments are well-defined (Wittlinger et al., 2004; Xiao et al., 2017; Xie et al., 2024; Wu et al., 2024; Yao et al., 2025), its eastern termination and continuation remain highly controversial due to extensive sedimentary cover of the desert and discontinuous bedrock outcrops (Yue et al., 2001; Yang et al., 2023). Definition of its eastern extent is critical for understanding deformation propagation into northeast Tibetan plateau and for regional seismic hazard assessment. Yue et al. (2001) proposed that the ATF likely extends east-northeast to northeast along the Alxa–East Mongolia fault (Donskaya et al., 2008), with Cenozoic motion displacing the Hexi Corridor and cutting through the BOC. An alternative view suggests that the ATF transfers its slip northward into the thrust systems along the northern margin of the Hexi Corridor, transitioning from predominantly strike-slip in the west to dip-slip in the east, where it eventually terminates against reverse faults within the Alxa Block (Xiao et al., 2016; Yang et al., 2023).

This study suggests that the ATF does not terminate at the Qilian Shan frontland, but rather extends eastward to east-northeast into the BOC and the Alxa Block. The notable velocity contrast observed across the fault F5 may indicate the ATF extends at least across the southern margin of the BOC. The projected left slip rate of 1.52–2.69 mm/a, is significantly higher than the thrust rate of 0.35 mm/a in the southernmost Beishan area (Yang et al. 2019; Yun et al. 2025). Geodetic velocities reveal that the moving direction changed from northeast in the Qilian Shan to nearly-east in the BOC with a significant lateral motion rates (Fig. 7; Yang et al., 2021). Darby et al. (2005) identified five major left -lateral strike-slip faults within the BOC and the Alxa block, which strike is consistent with the ATF, and accommodated ≥70 km post-Cretaceous offset based on Landsat/ASTER imagery and field mapping. Thus, we propose that the ATF continues along a consistent NEE strike at the southern margin of the BOC. Local faults identified within the southern margin of the BOC, such as the Beihewan fault, the Kuantanshan-Heishan fault, the Jiujing-Bantan fault, and the Ebomiao fault may be the secondary splays of the ATF and eventually terminate against it (Fig. 7; Zhang et al., 2021; Yang et al., 2019; 2024b; Yun et al., 2025). These subsidiary structures collectively accommodate distributed crustal deformation regionally. This kinematic model also provides a coherent explanation for the distinct slip-rate decline from the western-central to eastern segment of the ATF (Yang et al., 2024b). Furthermore, the distribution of seismicity supports this structural boundary: all recorded earthquakes of  $M \ge 4.7$  occur south of the ATF and its eastern prolongation, with no events detected north of the fault within the interior of the BOC (Fig. 7).

## Conclusion

- 1) Crustal thickness decreases northward from 60 km beneath the central Jiuquan basin to 47.5 km beneath the Que'ershan arc; average crustal (6.24–6.43 km s<sup>-1</sup>) and uppermost mantle velocities (7.7–8.1 km s<sup>-1</sup>) reveal strong lateral heterogeneity.
- 2) North-dipping velocity contours and a lower-crust-upper-mantle low-velocity corridor beneath the Hexi Basin support early Paleozoic north-dipping subduction of the Qilian Ocean; a positive upper-mantle anomaly (8.0–8.3 km s<sup>-1</sup>, 45–70 km) between faults F2 and F4 represents a

- fossil break-off slab after north-dipping subduction of the Beishan Ocean.
- 3) Fault F5 acts as a lithospheric-scale partition: north-vergent, decoupled thrusting south of F5 versus whole-crust gentle folding to the north; a dome-shaped Moho and slow mantle Pn phase (7.7–7.9 km s<sup>-1</sup>) beneath F5 localize present-day strain.
- 4) The ATF continues east-northeast across the fault F5 into the Beishan and Alxa Block; and secondary splays of the ATF along the southern Beishan margin collectively accommodate deformation and explain the eastward slip-rate decrease.

Acknowledgments This work is jointly funded by the National Natural Science Foundation of the China (Grant Nos. 42274134, 41774114, 42261144669), the Deep Earth Probe and Mineral Resources Exploration-National Science and Technology Major Project (2025ZD1007701), the China Geological Survey Project (Grant No. DD20230008, DD20179342 and DD20160083), and China Scholarship Council. We acknowledge the two anonymous reviewers and Professor Xu Tao for their constructive comments.

## Code/Data availability

The other seismic shot records are supplied as supplementary material.

#### **Author contribution**

XSX: Writing-original draft, Writing-review & editing, Visualization, Validation, Investigation, Data curation, Methodology, Formal analysis, Conceptualization, Project administration. YKL: Writing-draft writing, Investigation, Data curation, Methodology, Validation, Conceptualization. XHC: Funding acquisition, Conceptualization. GWW: Visualization, Methodology. RG: Supervision, Project administration. JDE: Writing – review & editing. All authors contributed to the article and approved the submitted version.

## **Competing interests**

The authors declare that they have no known competing financial interests or personal relationships that could have appeared to influence the work reported in this paper.

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

Table 1 Explosion position of the deep seismic sounding profile

| Shot No. | Longitude | Latitude | Elevation | TNT charge | Shot time    |  |
|----------|-----------|----------|-----------|------------|--------------|--|
|          | (°E)      | (°N)     | (m)       | (kg)       | (UTC+8)      |  |
| ZB0      | 98.406    | 38.825   | 3387      | 2000       | 00:00:04.900 |  |
| ZB1      | 97.761    | 39.702   | 2466      | 2000       | 23:59:46.890 |  |
| ZB2      | 97.724    | 40.074   | 1447      | 2000       | 23:01:44.285 |  |
| ZB3      | 97.965    | 40.512   | 1334      | 1500       | 22:04:11.157 |  |
| ZB4      | 98.271    | 41.038   | 1302      | 1500       | 22:01:44.340 |  |
| ZB5      | 98.367    | 41.495   | 1341      | 2000       | 22:00:13.350 |  |
| ZB6      | 98.364    | 41.937   | 1520      | 3000       | 23:00:27.270 |  |
| ZB7      | 98.818    | 42.172   | 1243      | 2000       | 20:05:02.438 |  |
| ZB8      | 98.958    | 42.387   | 1154      | 3000       | 23:03:05.596 |  |

| T-1-1- 2 TL- | DMC          | C 4 14 !       | - C:44: C-   | 1:004       | seismic phases |  |
|--------------|--------------|----------------|--------------|-------------|----------------|--|
| Table / The  | e KIVIS erro | r of fravellim | e titting to | r aitterent | seismic phases |  |

|                            | Pg                 |                    | P2                 |                    | Р3                 |                    | P4                 |                    | Pm                 |                    | Pn                 |                    |
|----------------------------|--------------------|--------------------|--------------------|--------------------|--------------------|--------------------|--------------------|--------------------|--------------------|--------------------|--------------------|--------------------|
| Shot                       | No.<br>of<br>picks | RMS<br>time<br>(s) |
| ZB0                        | 16                 | 0.0251             | 26                 | 0.0137             | 33                 | 0.0169             | 37                 | 0.0144             | 47                 | 0.0159             | 17                 | 0.0127             |
| ZB1                        | 58                 | 0.021              | 58                 | 0.02               | 57                 | 0.0268             | 57                 | 0.0207             | 75                 | 0.0248             | 26                 | 0.0265             |
| ZB2                        | 52                 | 0.0309             | 46                 | 0.032              | 46                 | 0.0238             | 55                 | 0.0244             | 51                 | 0.0266             | 17                 | 0.0275             |
| ZB3                        | 48                 | 0.0157             | 60                 | 0.0226             | 61                 | 0.0144             | 59                 | 0.0158             | 44                 | 0.0123             |                    |                    |
| ZB4                        | 35                 | 0.0274             | 39                 | 0.0216             | 36                 | 0.0218             | 39                 | 0.0351             | 35                 | 0.0206             |                    |                    |
| ZB5                        | 31                 | 0.0118             | 29                 | 0.0108             | 43                 | 0.0191             | 34                 | 0.0189             | 48                 | 0.0225             | 13                 | 0.0204             |
| ZB6                        | 33                 | 0.0228             | 36                 | 0.0155             | 37                 | 0.0254             | 41                 | 0.0204             | 61                 | 0.0218             | 19                 | 0.0256             |
| ZB7                        | 32                 | 0.0246             | 29                 | 0.019              | 31                 | 0.0221             | 33                 | 0.0197             | 48                 | 0.018              | 21                 | 0.0238             |
| ZB8                        | 28                 | 0.0309             | 25                 | 0.033              | 33                 | 0.0233             | 55                 | 0.0228             | 53                 | 0.0272             | 12                 | 0.0228             |
| Mean<br>RMS<br>time<br>(s) |                    | 0.0234             |                    | 0.0209             |                    | 0.0215             |                    | 0.0214             |                    | 0.0211             |                    | 0.0227             |
| Total<br>picks             | 333                |                    | 348                |                    | 377                |                    | 410                |                    | 462                |                    | 125                |                    |

Fig. 1 (a) Simplified tectonic framework of East Asia, emphasizing the CAOB and the Tethys tectonic belt (modified after Xiao et al. 2020). (b) Map of the distribution of fault system in the NE Tibetan Plateau, the Beishan Orogenic Collage and surrounding regions (modified from Zhang et al. 2023). Base map: SRTM 30 m DEM, NASA/USGS (public domain). Processing: Global Mapper v24.0. (c) Map of the main faults in the NE Tibetan Plateau, the Beishan Orogenic Collage, showing the location of the seismic profile, faults and ophiolite mélange zones (modified from Yin et al., 2008 and Xiao et al., 2010). F1: the Hongshishan-Baiyueshan-Pengboshan suture zone; F2: the Jijitai-Shibanjing-Xiaohuangshan fault; F3: the Hongliuhe-Baiyunshan-Yueyashan-Xichangjing suture zone; F4: the Liuyuan-Zhangfangshan suture zone; F5: the southern margin fault of the Beishan; F6: the Kuantanshan-Heishan fault; F7: the Yumen-Shuigoukou fault; F8: the northern margin fault of the North Qilian; F9: the North Qilian fault; F10: the Altyn Tagh fault. The digital topographic basemaps and profiles from GeoMapApp software (Ryan et al., 2009) are available at http://www.geomapapp.org/.

Fig. 2 P-wave record section (on a reduced time scale) corresponding to the shot point ZB1. (a) identified seismic phases; (b) velocity layers and ray tracing. The seismic phases are marked by dotted lines, and the squares mark the location of calculated traveltimes.

Fig. 3 P-wave record section (on a reduced time scale) corresponding to the shot point ZB8. (a) identified seismic phases; (b) velocity layers and ray tracing. The seismic phases are marked by dotted lines, and the squares indicate the location of calculated traveltimes.

Fig. 4 Traveltime fitting and ray coverage of all shots. (a) ray tracing coverage of the model; (b) rays of all the phases for nine shots.

Fig. 5 2-D crustal-upper mantle velocity structure. Upper section: topography showing main tectonic units; lower section: crustal-upper mantle velocity structure. White solid lines denote the main interfaces resolved from wide-angle reflections.

Fig. 6 2-D crustal- upper mantle velocity anomaly structure (layer means subtracted from velocities shown in Fig. 5). Upper section: topography showing the main tectonic units; lower section: crustal-upper mantle velocity anomaly structure. White solid lines denote the main interfaces resolved from wide-angle reflections.

Fig. 7 Earthquake distribution, GPS motion direction, and velocity in The NE Tibetan Plateau and the Beishan Orogenic Collage (modified from Yang et al., 2021). Purple arrows indicate GPS vectors relative to stable Siberia. The dashed white line indicates the proposed eastern extension of the ATF. BHWF: Beihewan Fault; JJF: Jiujing fault system; MZSF: Mazongshan fault system; SWSF: Sanweishan Fault; DS: Daquan Shan; YS: Yushi Shan.

Fig. 8 Tectonic interpretations of features derived from the velocity model across the North Qilian–Beishan Orogenic Collage. Blue dotted lines indicate the inferred subduction directions of the fossil subduction slabs. Red solid lines denote the faults imaged at depth, whereas the white solid and dashed lines represent surface fault traces.