# Peer review of "Crustal-upper mantle velocity structure from the North Qilian"

_EGUsphere, 2025_

## Referee Comment (RC1)

**General comment:**

This study presents the crust and upper mantle velocity structure extending from the North Qilian to the Beishan block and discusses the tectonic significance of the observed crustal deformation. The newly acquired dataset, a 460-km-long seismic wide-angle and refraction profile, appears to have been carefully collected and processed, and provides valuable insights into the deep lithospheric structure of the region. The manuscript would benefit from careful English editing to improve clarity and readability. In particular, some expressions are overly colloquial and should be revised to meet the conventions of scientific writing. I hope these comments are helpful and contribute to improving the overall quality of the manuscript.

**General comments:**

Q1: In lines 58–66 of the introduction, the text appears to summarize the main conclusions of the study. It may be more appropriate to move this content to the conclusion section

Q2: The manuscript states that the crustal-upper mantle structure remains ambiguous due to limited resolution. Could the authors clarify the actual resolution of the present data and indicate whether it is higher than in previous studies? Additionally, please specify which aspects remain unresolved and how this study's findings differ from prior work.

Q3: Please note that in scientific writing, en dashes (–) rather than hyphens (-) should be used to indicate numerical ranges (e.g., 0.3–1.0 km/s). Please pay attention to the use of definite articles (e.g., 'the') to improve grammatical accuracy. Additionally, check the capitalization of all proper nouns, including geographic names, tectonic units, and geological terms, and maintain consistency throughout the manuscript.

Q4: In the "Crustal Velocity Structure Implications" parts, how does this velocity value inform the structure implications? Providing explicit links between the velocity data and geological implications would strengthen this section.

Q5: "The crustal velocity structure proposes an unusual scenario where the deepest Moho is found in the central Jiuquan basin, rather than the North Qilian Shan with the highest elevation. Could you explain it in the manuscript?

Q6: The conclusion section currently shows formatting inconsistencies and incorrect numbering. A careful revision is recommended. Furthermore, restructuring the conclusions to more clearly highlight the key scientific findings would enhance the clarity and impact of this section.

Q7: It is suggested that the formatting of both in-text citations and the reference list be revised and standardized to ensure consistency with the journal's guidelines.

**Detailed comments and corrections:**

Line 21: "five strata" → "five layers"

Line 35: Before using the simplified CAOB, it's better to add it in Line 35 after the "Central Asian Orogenic Belt".

Line 42: Figure 1b → Fig. 1b

Line 61: Removing the excess space before "Notably".

Line 69: "In Cenozoic" → In the Cenozoic or during the Cenozoic.

Line 73: "of NE Tibet" → "of the NE Tibet"; please check and correct similar expressions throughout the manuscript.

Line 78: Removing the excess space.

Line 80: HUANG et al. 2014 → Huang et al. 2014

Line 82: a N-S-trending → an N-S-treading

Line 96: Delete "respectively".

Line 99: Please clarify the meaning of "the final sealing position."

Line100:  North Beishan block → North Beishan Block; in middle-late Ordovician>> in the Middle to Late Ordovician

Line 131: What's the meaning of "TNT"?

Line 147: the travetime of ZB1→ The travetime of ZB1

Line 159: Delete repeated parentheses.

Lines 168–172: P1–P4 are not shown in Fig. 5; please clarify or adjust the text accordingly.

Lines 239–244: Specify which figure corresponds to this phase.

Line 254: Text formatting is inconsistent; please revise.

Line 258: -1.1–-0.15 km/s → -1.1 – -0.15 km/s or "ranges from -1.1 to -0.15 km/s"

Line 281: The text formatting is not standard.

Line 310: Consider deleting the semicolon (";") and revising lines 310–313 for clarity and grammar.

Line 345: The abbreviation "Mts." is informal; use "Mountains" instead. Line 347: The comma should be deleted.

Line 371: How is the decoupled crust inferred from the seismic profile in this study? Or is this based on previous studies? Please clarify.

Figure 5&6: Letters (a) and (b) are not shown on the figures. It is suggested to mark the north (N) and south (S) directions for clarity.

---

## Referee Comment (RC2)

The authors here focus on the P wave velocity structure in a significant region of the north-east Tibetan Plateau and the southern segment of the Central Asian Orogenic Belt, using seismic wide-angle reflection and refraction profile. The data is precious here, and the velocity structure can be the key for us to understand the north-east expansion of the Tibet and the tectonic process of the Paleo-Asian oceanic.

The data process and uncertainty analysis for the inversion of velocity structure are detailed and reliable. However, the manuscripts have a large problem with writing. Many sentences are expressed vaguely and do not conform to grammar rules. The authors need to improve their English writing, so that they can make their interpretation clarity.

I'm inclined to suggest that this paper be published after the authors revise the English writing and all the questions as follows.

General Comments:

Q1: Please use consistent abbreviations and use the full spelling for the first occurrence of an abbreviation, e.g. CAOB, PAO. And make all the units be uniform, for example, the authors first use "km" and then use "kilometers".

Q2: I think the authors use ZPLOT to pick the arrivals and apply RAYINVR to get the velocity structure. However, they didn't mention the software in the text. I cannot rule out the possibility that they used other

methods. If so, please add them in the methods section.

Q3: What's the uncertainty when they picked the refraction and reflection arrivals?

Q4: Generally, the P wave velocity of upper crust is <6.4 km/s, middle crust is 6.4-6.8 km/s and lower crust is >6.8 km/s. The authors stated that they divided the upper crust from surface toP1, the middle crust from P2 to P3 and the lower crust from P3 to the Moho discontinuity (Line 169-172). What's the refer for their stratification? It's clear that the layer above the Moho is lower crust, which velocity is ~6.8km/s. If they make the P4 to the Moho as lower crust, their statements for the co-thickening of the middle-lower crust should be middle crustal thickening.

Q5: The velocity value is reliable in the region where have ray coverage. The authors should make it clarity for the resolved and unresolved velocity region. They stated that the Pn velocity in the upper mantle is ~7.7-8.6km/s. In my eyes, the Pn velocity is correct in the regions where have Pn's ray coverage, the max Pn velocity they can constrain is no more than 8.3km/s.

Q6: I do not think the authors have enough evidence for the conclusion that the upper crust is decoupled with the middle-lower crust (Line 371-373). If they got the conclusion based on previous studies, they should give robust analysis.

Q7: The authors stated that F5 is a regional large-scale strike-slip fault. This contribution is crucial significance for understanding the tectonic

mechanism between the CAOB and the Qinghai-Tibet Plateau. Can they give robust evidence to explain how this regional fault reconciles the huge displacement differences on both sides of the fault? Why are there no deep earthquakes along the local fault? This is very important for their conclusion.

Detailed Comments:

Line 19: "seismic wide angle and refraction profile spanning the……",

Incorrect usage of professional terms, "seismic wide angle and refraction profile" must be "wide angle reflection and refraction profile".

"Spanning the……" should be "spanning from the……".

Line 20-21; 36-38; 58-59 etc. These sentences are ambiguous; a native English editing is required.

Line 39: "CAOB"

When an abbreviation is first used, its full form should be used.

Line 61-62: "Notably, … inhomogeneity non the …,".

It is a mistake for "non".

Line 83-84: "…… refraction profile sweeps throughout the North Qilian, Hexi corridor (containing the Jiuquan basin and the Huahai basin), and the entire Beishan block was done".

Two predicates (sweep and was) are used in a single sentence.

Line 138-139: "To make the seismic records clearer, each trace was

bandpass filtered up to 8 Hz……".

It is vague for the meaning of this sentence. The authors could write like "To improve the signal-to-noise ratio, we apply bandpass filter …... from … Hz to … Hz……".

Line 177: "greater velocity zone"

Higher velocity zone

Line 186: "with an interface depth falling to 11.2–12.5 km."

"Falling" is very strange here.

Line 188: "a high-velocity body".

High-velocity zone or high velocity abnormality will be a better choice.

Line 200: "interface depth climbs to 17.6–27.5 km".

"Climbs" is very strange here.

Line 201-203: "This characteristic shows that the North Qilian and the Jiuquan basin have a consistent basement, matching with the residual gravity anomaly findings (Yang et al. 2024).".

According to the interpretation from the authors: there is a high velocity zone ~10 km below the North of Qilian, the velocities are totally different when compared to Qilian and Jiuquan basin. How did they get the conclusion that the North Qilian and the Jiuquan basin have a consistent basement?

Line 209: "the Jiuquan basin is 23.4–38.7 kilometers".

It is necessary to keep consistency for the depth unit, e.g. using "km" in

the whole text.

Line 215: "The interval velocity increases to 6.3–6.42 km/s".

Which part of the profiles show the velocity increases to 6.3-6.4 km/s?

Line 239-244: In this part, the authors try to state the difference features beneath the central part of the profiles. However, they should use more precise interpretation when using Pn velocity which is resolved by ray coverage. According to the ray coverage, the Pn velocity is not as high as they declared 8.4-8.6 km/s.

Line 293-295: "…… (0.01 - 0.1) …… (-0.01 - -0.12)".

The authors missed the velocity units "km/s".

Line 311: "while past geophysical ……".

It is much better to write "while previous studies ……".

Line 331-332: "…… the crust north of the Que'ershan subducted ……".

Such a sentence structure is obviously incorrect.

Line 394 and 402: the authors forgot the numbers (1) and (3).

Line 409-411: the authors should complete the sentence, and make it correct.

Fig.1b and c: remove the faults which are not discussed in the manuscripts. It looks Indistinguishable and chaotic.

Fig. 2 and Fig. 3: To make the figures clarity, the authors should adjust these two figures to be the same size. And I suggest the authors add a white background to the letters (a) and (b).

Fig.5 and Fig.6: adding (a) and (b) on the correct profiles, marking the direction "SW" and "NE", and giving the region of resolved and unresolved velocity according to the ray coverage.

---

## Author Comment (AC4)

This study presents the crust and upper mantle velocity structure extending from the North Qilian to the Beishan block and discusses the tectonic significance of the observed crustal deformation. The newly acquired dataset, a 460-km-long seismic wide-angle and refraction profile, appears to have been carefully collected and processed, and provides valuable insights into the deep lithospheric structure of the region. The manuscript would benefit from careful English editing to improve clarity and readability. In particular, some expressions are overly colloquial and should be revised to meet the conventions of scientific writing. I hope these comments are helpful and contribute to improving the overall quality of the manuscript.

**Response to Reviewer Comments**

We thank the reviewer for the thorough review and constructive comments, which have significantly helped improve the quality of our manuscript. We have carefully considered all points raised. Our point-by-point responses and the planned revisions are detailed below.

We acknowledge the comment regarding the need for careful English editing. We thoroughly revised the entire manuscript to improve clarity, readability, and adherence to the conventions of scientific writing. This includes correcting colloquial expressions, improving grammatical accuracy, and ensuring a formal tone throughout.

**Q1:** In lines 58–66 of the introduction, the text appears to summarize the main conclusions of the study. It may be more appropriate to move this content to the conclusion section.

**Response:** Agreed. We reorganized the "Introduction" section (lines 48-95), and deleted the summary of the main conclusions from the introduction (lines 95-143). The introduction was revised to maintain its focus on presenting the research problem, context, and objectives.

**Q2:** The manuscript states that the crustal-upper mantle structure remains ambiguous due to limited resolution. Could the authors clarify the actual resolution of the present data and indicate whether it is higher than in previous studies? Additionally, please specify which aspects remain unresolved and how this study's findings differ from prior work.

**Response:** Our data provides higher resolution of velocity structure in the same study area. Compared to the "Golmud-Ejin" wide-angle reflection and refraction profile acquired in 1992, we used dense shot interval and station spacing, and higher yield explosive. The geophones we used are much more sensitive to the seismic waves than the ones before (the detailed parameters are shown as follows).

The sentences "Nevertheless, the crustal-upper mantle velocity structure remains ambiguous because of the wide shot and receiver spacing and the relatively weak

explosive sources inherited from the earlier, lower-resolution acquisition design (Cui et al., 1995)." Explained the reason (lines 85-88)

| Seismic profile | Shot interval | Station spacing | TNT | Record medium |
|---|---|---|---|---|
| Golmud-Ejin | 80-200 km | 4 km | 1.5 T | Tape |
| This study | 40-60 km | 2-3 km | 1.5-3 T | Digital |

**Q3:** Please note that in scientific writing, en dashes (-) rather than hyphens (-) should be used to indicate numerical ranges (e.g., 0.3–1.0 km/s). Please pay attention to the use of definite articles (e.g., 'the') to improve grammatical accuracy. Additionally, check the capitalization of all proper nouns, including geographic names, tectonic units, and geological terms, and maintain consistency throughout the manuscript.

**Response:** We performed a thorough check and correction of the entire manuscript to: 1) replace all hyphens with en dashes in numerical ranges, 2) carefully review and correct the use of definite articles ('the') for grammatical accuracy, and 3) standardize the capitalization of all proper nouns and ensure consistency throughout the text. (line 406)

**Q4:** In the "Crustal Velocity Structure Implications" parts, how does this velocity value inform the structure implications? Providing explicit links between the velocity data and geological implications would strengthen this section.

**Response:** We considered that placing the "Crustal Velocity Structure Implications" between the "Velocity Structure" and "Discussion" sections was somewhat structurally unconventional. To improve the logical flow of the manuscript, this subsection has been integrated into the "Introduction" section. (line 393-397)

**Q5:** "The crustal velocity structure proposes an unusual scenario where the deepest Moho is found in the central Jiuquan basin, rather than the North Qilian Shan with the highest elevation. Could you explain it in the manuscript?

**Response:** Yes, we have carefully considered this observation. We propose that the North Qilian Shan and the Jiuquan Basin share a common basement, which explains their comparable Moho depths. Although the Moho beneath the Jiuquan Basin is slightly deeper, the North Qilian Shan exhibits a higher surface elevation, indicating a significantly thicker crust overall when topographic compensation is taken into account.

The sentence "These characteristics indicate that the NQS and the Jiuquan basin share a consistent basement structure, which aligns with the findings from residual gravity anomaly analyses (Yang et al. 2024)." explained the reason. (lines 330-333)

**Q6:** The conclusion section currently shows formatting inconsistencies and incorrect numbering. A careful revision is recommended. Furthermore, restructuring the conclusions to more clearly highlight the key scientific findings would enhance the clarity and impact of this section.

**Response:** We carefully reformatted the conclusion section to correct numbering and formatting. We also restructured it to concisely and clearly list the key scientific findings first, followed by their broader implications, thereby enhancing the section's clarity and impact. (lines 667-680)

**Q7:** It is suggested that the formatting of both in-text citations and the reference list be revised and standardized to ensure consistency with the journal's guidelines.

**Response:** We meticulously revised the formatting of all in-text citations and the reference list to ensure they are complete and fully consistent with the specific guidelines of the target journal. (Line 706)

**Detailed Comments and Corrections:**

- **Line 21: "five strata" → "five layers"**

    "five strata" was changed to **"five layers"**. (Line 25)

- **Line 35: Before using the simplified CAOB, it's better to add it in Line 35 after the "Central Asian Orogenic Belt".**

    The simplified acronym **CAOB** was added in parentheses after its first full mention: **"Central Asian Orogenic Belt (CAOB)"**. (line 55)

- **Line 42: Figure 1b → Fig. 1b**

    "Figure 1b" was changed to **"Fig. 1b"** (and consistency for all figure citations was checked). (Line 160)

- **Line 61: Removing the excess space before "Notably".**

    We reorganized the introduction; the sentence was deleted. (line 116)

- **Line 69: "In Cenozoic" → In the Cenozoic or during the Cenozoic.**

    "In Cenozoic" was changed to **"In the Cenozoic"**. (Line 155, line 453)

- **Line 73: "of NE Tibet" → "of the NE Tibet"; please check and correct similar expressions throughout the manuscript.**

    "of NE Tibet" was changed to **"of the NE Tibetan Plateau"** (and similar expressions throughout the manuscript were corrected). (Line 65, line 71, line 74, line 449)

- Line 78: Removing the excess space.

    We reorganized the introduction, the sentence was deleted. (lines 133-143)

- **Line 80: HUANG et al. 2014 → Huang et al. 2014**

    "HUANG et al. 2014" was changed to **"Huang et al., 2014"** (and citation formatting was standardized). (Line 84 )

- **Line 82: a N-S-trending → an N-S-treading**

    "a N-S-trending" was changed to **"SW-NE-trending"**. (line 88 )

- **Line 96: Delete "respectively".**

    The word **"respectively"** was deleted as suggested (line 177).

- **Line 99: Please clarify the meaning of "the final sealing position."**

The term **"the final sealing position"** was changed to "**amalgamation position**" to express the eaxt meaning. (line 116 )

- **Line100:North Beishan block → North Beishan Block; in middle-late Ordovician>> in the Middle to Late Ordovician.**

 "North Beishan block" was capitalized to **"North BOC"**. (line 181)

"in middle-late Ordovician" was changed to **"in the middle to Late Ordovician"** (line 182).

- **Line 131 What's the meaning of "TNT"?**

The meaning of the acronym **"TNT"** was refered to "**Trinitrotoluene**", which was upon its first use in the manuscript in line 217.

- **the travetime of ZB1→ The travetime of ZB1**

We reorganized this sentence, which now is "**The travetimes recorded at shotpoint ZB1**…" ((line 244-245)

- **Line 159: Delete repeated parentheses.**

The repeated parentheses were deleted (Lines 259-260)

- **Lines 168–172: P1–P4 are not shown in Fig. 5; please clarify or adjust the text accordingly.**

**P1-P4 are actually C1-C4, we corrected this mistake**. The revised sentences are "The typical continental crust is stratified into three principal layers: the upper crust, comprising sedimentary cover overlying crystalline basement characterized by an average P-wave velocity of 6.0–6.3 km s$^{-1}$; the mid-crust, composed of interleaved silicic and basic lithologies, with velocities of 6.3–6.5 km s$^{-1}$; and the lower crust, dominated by more mafic assemblages, exhibiting velocities of 6.6–6.9 km s$^{-1}$ (Christensen, 1995; Jia et al., 2019). Based on our velocity structure result, the crust can be divided into upper crust (from the surface to C2), middle crust (from C2 to C3), and lower crust (from C3 to the Moho). The upper crust can be separated into two layers by intracrustal interface C1 determined by seismic phase P2. The lower crust can also be subdivided into two layers by intracrustal interface C4 indicated by seismic phase P4." (line 282-286)

- **Lines 239–244: Specify which figure corresponds to this phase.**

We put the figure 5 behind the sentence, which is as "The upper mantle velocity structure exhibits distinct lateral variations across the study area (Fig. 5). (line 382)

- **Line 254: Text formatting is inconsistent; please revise.**

  The sentence was revised as "To improve the visibility of the velocity heterogeneity of the crustal-upper mantle structure the mean layer velocities are subtracted to produce a velocity anomaly structure of the crustal-upper mantle (Fig. 6). "(lines 399-403)

- **Line 258: -1.1–0.15 km/s → -1.1 – -0.15 km/s or "ranges from -1.1 to -0.15 km/s"**

  **Line 258:** "-1.1—0.15 km/s" was corrected to **"-1.1– -0.15 km s$^{-1}$"**. (line 406)

- **Line 281: The text formatting is not standard.**

  The sentence was revised as "The BOC is characterized by a strong positive velocity anomaly (0.02–0.12 km s$^{-1}$) and exhibits a northward-increasing trend (Fig. 6)." (line 428-429)

- **Line 310: Consider deleting the semicolon (";") and revising lines 310–313 for clarity and grammar.**

  The sentences were revised as "Although previous geophysical investigations have covered the Qilian Shan (Xiao et al. 2016; Guo et al. 2019; Shen et al. 2020; Li et al. 2021), focus has largely been on the neotectonics rather than the Paleozoic evolution. In this study, we observed north-dipping velocity contour from interface C2 to the uppermost mantle beneath the Qilian Shan, coupled with a lower-crust–upper-mantle low velocity anomaly beneath the Hexi Corridor (Fig. 6). These features most plausibly record early Paleozoic north-dipping subduction of the Qilian Ocean, which aligns with the surface geology; later collisional or bidirectional shortening may have locally overprinted the original polarity (Davis and Darby, 2010)." (line 484-491)

- **Line 345: The abbreviation "Mts." is informal; use "Mountains" instead. Line 347: The comma should be deleted.**

  The informal abbreviation **"Mts."** was replaced with the full word **"mountain belt"**. (line 476)

- Line 347: The comma should be deleted.

  We have thoroughly reviewed the manuscript and made corrections to all punctuation errors throughout.

- **Line 371: How is the decoupled crust inferred from the seismic profile in this study? Or is this based on previous studies? Please clarify.**

  The **decoupled crust** was interpreted based on our seismic profile, the base of interface C1 is acting as the decollement as shown in the Fig. 5. ("correlate to

the north-vergent thrusting that decouples from a co-thickened middle-lower crust (25–50 km, 6.5–6.8 km s⁻¹), which is consistent with the reflection structure (Xiong et al., 2025)." (lines 518-520)

- Figure 5 & 6: Letters (a) and (b) are not shown on the figures. It is suggested to mark the north (N) and south (S) directions for clarity.

Given that Figures 5a and 6a are not referenced in the main text, we have omitted the subplot labels (a) and (b) from Figures 5 and 6. Accordingly, all citations of these figures in the manuscript have been updated from "Fig. 5b" to "Fig. 5" and from "Fig. 6b" to "Fig. 6" to ensure consistency. **Northeast (NE) and Southwest (SW) directional indicators** was clearly marked on the figures for orientation. (line 1103, line 1107)

---

## Author Comment (AC5)

**General Comment:**

This manuscript presents a detailed investigation of the crustal-upper mantle velocity structure across the North Qilian Shan to the Beishan block using a 460-km-long seismic wide-angle reflection/refraction profile. The study provides valuable insights into the tectonic evolution of the northeastern Tibetan Plateau and the southern Central Asian Orogenic Belt (CAOB). The seismic profile is well-designed, and the processing techniques (e.g., phase identification, velocity modeling) are appropriately applied. The error analysis (e.g., RMS traveltime residuals) supports the reliability of the results. The proposed northward subduction polarity of the Qilian Ocean and the role of the southern Beishan boundary fault (F5) as a major strike-slip structure are significant contributions. The findings enhance understanding of crustal deformation mechanisms in the transition zone between the Tibetan Plateau and the CAOB. The data are robust, and the methodology is sound, but the manuscript requires improvements in clarity, interpretation, and presentation before it can be published finally:

**Response:**

We extend our sincere thanks to Prof. Xu for his positive evaluation of our work and for providing valuable suggestions. In response, we have undertaken a comprehensive revision of the manuscript aimed at enhancing its clarity, interpretive depth, and overall presentation. Specifically, we have rephrased both the Introduction and Discussion sections to improve logical coherence and scientific rigor. The detailed responses to each comment are provided below.
* * *
**Q1: Terminology Consistency: Use either "Beishan block" or "Beishan orogenic belt" consistently. Define abbreviations (e.g., PAO, CAOB) at first use.**

**Response:**

We have standardized the terminology throughout the manuscript, using **"Beishan orogenic collage"** (BOC, line 23) consistently, as it accurately reflects the complex accretionary nature of the region. All abbreviations, including **P**aleo-**A**sian **O**cean (PAO, line 58) and **C**entral **A**sian **O**rogenic **B**elt (CAOB), are now explicitly defined upon their first occurrence in the text The simplified acronym CAOB was added in parentheses after its first full mention: "Central Asian Orogenic Belt (CAOB)". (line 55).
* * *
**Q2: In lines 33–34 of the introduction, the sentences are overly complex or ambiguous; a thorough language edit by a native English speaker is recommended.**

**Response:**

We have performed a comprehensive language edit of the entire manuscript with the assistance of a native English speaker. The sentences in lines 33–34 and similar complex passages throughout the introduction have been simplified and rewritten for improved clarity and readability. the original text of lines 33-34 has been revised to: " The NW-SE-trending Qilian Shan, situated in the NE Tibet, is bounded by the Altyn Tagh fault (ATF) to the west, the

northern Qaidam thrust system to the south, the Haiyuan fault to the east, and the north Qilian Shan fault to the north (Fig. 1b). The present-day Qilian Shan exhibits a Cenozoic fold-thrust belt with multi-stage tectonic deformation prior to the Cenozoic (Yin and Harrison 2000; Gehrels et al. 2003; Song et al. 2014; Wu et al. 2016; Zuza et al. 2017; 2019). North of the Qilian Shan, the Hexi corridor basins and the Beishan Orogenic Collage (BOC) form the southern section of the Central Asian Orogenic Belt (CAOB, Xiao et al., 2010; Li et al., 2023; Xiong et al., 2024)." **(line 48-56)**

The whole Introduction section was reorganized in line 48-95.
* * *
**Q3: For figure captions: Fig. 1: Add scale bars and clarify tectonic unit labels; Fig. 5-6: Improve visibility of velocity contours and annotations.**

**Response:**
We have revised all figures as suggested:

- **Fig. 1:** Scale bars have been added, and all tectonic unit labels have been clarified and made consistent with the text (line 1072).

- **Fig. 5 & 6:** The visibility of velocity contours and annotations has been enhanced by adjusting line weights, colors, and font sizes. Poorly-resolved areas based on ray coverage have been masked to prevent overinterpretation (line 1103, line 1107).
* * *
**Q4: Compare results with existing seismic/gravity/MT studies (e.g., Cui et al., 1995; Xiao et al., 2017) to strengthen interpretations. Discuss potential biases (e.g., ray coverage gaps, trade-offs between velocity and interface depth).**

**Response:**
A new subsection has been added to the **Discussion Section** to compare our findings with existing geophysical studies:

- Our velocity model is now compared with results from Cui et al. (1995), Xiao et al. (2015), and other key seismic, gravity, and magnetotelluric (MT) models. This comparison strengthens our interpretations of crustal nature of the Beishan Orogenic Collage (line 459-470).

- We explicitly discuss potential biases and limitations, including **ray-density gaps** and the velocity–interface trade-off in seismic inversion. (line 271-276)
* * *
**Q5: In the "Discussion" section, compare results with existing seismic/gravity/MT studies (e.g., Cui et al., 1995; Xiao et al., 2017) to strengthen interpretations. Provide more geological evidence (e.g., paleo-trench positions, slab remnants) to support the north-dipping Qilian Ocean model.**

**Response:**

As noted in Q4, we have expanded the **Discussion section** to include direct comparisons with previous geophysical studies. Furthermore, we have integrated additional **geological evidence** to support the north-dipping subduction model for the Qilian Ocean:

- The positive upper-mantle velocity anomaly we identify is discussed as a potential **slab remnant**, linking it to the north-dipping subduction and closure of the Beishan Ocean. This provides a more robust, multi-disciplinary foundation for our tectonic interpretations (line 484-506).
* * *
**Q6: Update citations (e.g., Wu et al., 2024; Xie et al., 2023; Yao et al., 2025; Zhang et al., 2023) and include key regional studies (e.g., Zuza et al., 2019).**

**Response:**

The reference list has been thoroughly updated to include the suggested recent publications (Wu et al., 2024; Xie et al., 2023; Zhang et al., 2023,in line 609-611) and key regional studies (Zuza et al., 2019; in line 157, 480,540). All in-text citations have been checked for consistency and relevance. The reference list now comprehensively reflects the current state of knowledge in the field.
* * *
**Detailed Comments and Corrections:**

- **Line 1-3 (Title):** Modified to: **"Crustal-Upper Mantle Velocity Structure from the North Qilian Shan to the Beishan Block: Tectonic Significance of Crustal Deformation"**.

  **We adopted this suggestion, the title was changed to "**Crustal-upper mantle velocity structure from the North Qilian Shan to Beishan Orogenic Collage: tectonic significance of crustal deformation**" (line 2-3)**

- **Line 4-10 (Address):** Corrected extra commas (e.g., "Beijing 100094, China").

  **We corrected it.** (line 7-8)

- **Line 14:** "constitutes" → **"represents"**.

  **We corrected it.** (line 16)

- **Line 17:** "serves" → **"acts"**.

- **We rephrased this sentence as "**The crustal-mantle structure of the study area, a transition zone, is crucial to understanding the deep processes of accretion and crustal deformation.**" (line 16)**

- **Line 23:** "considerable variance" → **"significant variations (6.24–6.43 km/s)"**.

**We rephrased this sentence as** "Average crustal velocities (6.24−6.43 km s$^{-1}$) and Pn values (7.7−8.1 km s$^{-1}$) reveal strong lateral heterogeneity." **(line 28-29)**

- **Line 35–36:** Rewritten for clarity: **"As a transition zone between the NE Tibetan Plateau and the CAOB, the crust-mantle structure of the study area is crucial for understanding..."**

the crustal-mantle structure of the study area is crucial for understanding the regional evolution and interaction of Tibetan Plateau, part of the Tethys tectonic domain, and the PAO tectonic domain since the Paleozoic (Fig. 1a; Li et al. 1982; Yin and Harrison 2000; Xiao et al. 2009; Zhao et al. 2018; Xiong et al., 2024; He et al., 2025). (line 76-81)

- Line 43: "has witnessed" → "experienced"

We reorganized the Introduction section as the reviewers suggested; this sentence was deleted. (line 98)

- **Line 48:** "Experiencing multi-stage breakup..." → **"The block underwent multi-stage breakup..."**

- We reorganized the Introduction section as the reviewers suggested; this sentence was deleted. (line 102)

- **Line 138:** "To make... clearer" → **" To improve the signal-to-noise ratio, we applied each trace was bandpass filtered up to 8 Hz and displayed..."**

Line 138-139: Revised to: " Using the ZPLOT plotting package (Zelt, 1994), we performed trace editing, automatic gain control, band-pass filtering, velocity reduction, and phase picking for each shot. To improve the signal-to-noise ratio, we applied bandpass filter up to 8 Hz and displayed the seismic sections using a reduction velocity of 6 km s$^{-1}$ over a time window of -5–10 s (e.g. Fig. 2, Fig. 3).."(line 225-229)

- **Line 141:** "first arriving phase" → **"first-arrival phase"**.

It was corrected. (line 238)

- **Line 179:** "The base of interface C1 corresponds..." → **"Interface C1 marks the basement surface (3.4–6.5 km/s)..."**

It was corrected. (line 296)

- **Line 202:** "Qilian and the Jiuquan basin" → **"the NQS and the Jiuquan basin"**.

- It was corrected. (line 352)

- **Line 257-258:** "are with negative" → **"show negative"**.

- It was corrected. (line 406)

- **Line 259:** "which are prevented by" → **"which terminate at"**.

  It was corrected. (line 407)

- **Line 264:** "dives northward" → **"extends northward"**.

  It was corrected. (line 413)

- **Line 267:** "C1and C3" → **"C1 and C3"**.

- It was corrected. (line 416)

- **Line 313:** "Our data demonstrates" → "Our data demonstrate"

  The Discussion section was rephrased. This sentence was rewritten as "we observed north-dipping velocity contour from interface C2 to the uppermost mantle beneath the Qilian Shan, coupled with a lower-crust–upper-mantle low velocity anomaly beneath the Hexi Corridor…" (line 486)

- **Line 315:** "We speculate that" → "We interpret this as"

  The Discussion section was rephrased; we rewrote this sentence as '' These features most plausibly record early Paleozoic north-dipping subduction of the Qilian Ocean…"(line 488-489).

- **Line 338:** "could represent" → **"likely represents"**.

  It was corrected. (line 502)

- **Line 345:** "regarded as the youngest uplifted Mts." → "considered the most recently uplifted mountains"

  The Discussion section was rephrased; this sentence was deleted. (line 584)

- **Line 347:** "by a series" → **"through a series"**.

- The Discussion section was rephrased; this sentence was deleted. (line 585)

- **Line 381:** "was playing the function" → "functioned as"

  The Discussion section was rephrased; this sentence was deleted. (line 654)

- **Line 399:** "with the highest height" → "with the highest elevation"

  The Conclusion section was rephrased; this sentence was deleted. (line 686)

- **Line 402:** "is grouped" → "can be divided"

  The Conclusion section was rephrased; this sentence was deleted. (line 689)

- **Line 406:** "(3) Subduction…" → "Third, subduction…"

  The Conclusion section was rephrased; the numbering was reshaped. (line 667-680)

- **Line 408:** "(4) Bounded by..." → "Fourth, the F5 fault demarcates..."

  As the last question, the Conclusion section was rephrased; the numbering was reshaped. (line 667-680)

---

## Author Comment (AC6)

The authors here focus on the P wave velocity structure in a significant region of the north-east Tibetan Plateau and the southern segment of the Central Asian Orogenic Belt, using seismic wide-angle reflection and refraction profile. The data is precious here, and the velocity structure can be the key for us to understand the north-east expansion of the Tibet and the tectonic process of the Paleo-Asian oceanic.

The data process and uncertainty analysis for the inversion of velocity structure are detailed and reliable. However, the manuscripts have a large problem with writing. Many sentences are expressed vaguely and do not conform to grammar rules. The authors need to improve their English writing, so that they can make their interpretation clarity.

I'm inclined to suggest that this paper be published after the authors revise the English writing and all the questions as follows.

We sincerely thank the reviewer for the time and insightful comments on our manuscript. We have carefully considered all the suggestions and have revised the manuscript accordingly. Below is our point-by-point response.

We sincerely apologize for the language issues. The manuscript has now undergone comprehensive professional English editing to address vagueness, grammatical errors, and improve overall clarity and flow. We have also asked a native English-speaking colleague to proofread the revised version to ensure it meets the standards of scientific publication.
* * *
**Q1:** Please use consistent abbreviations and use the full spelling for the first occurrence of an abbreviation, e.g. CAOB, PAO. And make all the units be uniform, for example, the authors first use "km" and then use "kilometers".

**Response:**

Thank you for this important reminder. We have now ensured that all abbreviations are defined at first use, such as **C**entral **A**sian **O**rogenic **B**elt (**CAOB, line 55** ) , **P**aleo-**A**sian **O**cean (**PAO, line 58**). We have also standardized units throughout the manuscript, using **"km" (line 342)** and **"km s⁻¹" (line 32 etc.)** consistently, and have removed all instances of "kilometers" (line 342).
* * *
**Q2:** I think the authors use ZPLOT to pick the arrivals and apply RAYINVR to get the velocity structure. However, they didn't mention the software in the text. I cannot rule out the possibility that they used other methods. If so, please add them in the methods section.

**Response:**

The reviewer is correct. We have updated the Methods section to explicitly state the software used: " Using the ZPLOT plotting package (Zelt, 1994), we performed trace editing, automatic gain control, band-pass filtering, velocity reduction, and phase picking for each shot. (line 225-226).

"Model construction and editing are carried out with the RAYINVR software (Zelt and Smith, 1992)." (line 262-263)

We have also added the corresponding references to the reference list.
* * *
**Q3:** What's the uncertainty when they picked the refraction and reflection arrivals?

**Response:**
We have added a dedicated paragraph in the Methods section to quantify the picking uncertainty: " Uncertainties in phase picking primarily arise from challenging signal-to-noise conditions and complex subsurface wave propagation effects. The extensive desert sedimentary cover in the study area significantly attenuates seismic energy, particularly at larger offsets and for deeper arrivals. Additionally, strong lateral heterogeneities, such as fault zones and intracrustal velocity variation, cause substantial wave scattering, dispersion, and multipathing. This results in phase superposition and waveform distortion that complicates accurate phase identification. " (line 230-235)
* * *
**Q4:** …What's the refer for their stratification? It's clear that the layer above the Moho is lower crust, which velocity is ~6.8km/s. If they make the P4 to the Moho as lower crust, their statements for the co-thickening of the middle-lower crust should be middle crustal thickening.

**Response:**
We thank the reviewer for this insightful comment. According to Christensen, 1995 and Jia et al., 2019, the typical continental crust is stratified into three principal layers: the upper crust, comprising sedimentary cover overlying crystalline basement characterized by an average P-wave velocity of 6.0–6.3 km s⁻¹; the mid-crust, composed of interleaved silicic and basic lithologies, with velocities of 6.3–6.5 km s⁻¹; and the lower crust, dominated by more mafic assemblages, exhibiting velocities of 6.6–6.9 km s⁻¹. We thought that our stratification based on the presence of continuous seismic phases (P1, P2, P3, P4) which denote major intra-crustal interfaces is consistent with the previous wide-angle reflection and refraction profile across the Qilian and Alxa block east of our profile (Jia et al., 2019). Therefore, we add the sentences as follows to explain the reason for our crustal stratification.

"The typical continental crust is stratified into three principal layers: the upper crust, comprising sedimentary cover overlying crystalline basement characterized by an average P-wave velocity of 6.0–6.3 km s⁻¹; the mid-crust, composed of interleaved silicic and basic lithologies, with velocities of 6.3–6.5 km s⁻¹; and the lower crust, dominated by more mafic assemblages, exhibiting velocities of 6.6–6.9 km s⁻¹ (Christensen, 1995; Jia et al., 2019). Based on our velocity structure result, The result shows the crust can be divided into upper crust (from the surface to C2), middle crust (from C2 to C3), and lower crust (from C3 to the Moho)." (lines 277-282)

Thank you for this insightful comment. In response, we have refined our discussion on crustal thickening. We interpret that the uppermost crust is decoupled from the underlying crust. However, north of fault F5, the consistent undulation of interfaces from C2 down to the Moho suggests coherent deformation throughout the middle and lower crust. Therefore, we consider the term "middle-lower crustal co-thickening" to remain appropriate for describing

this region (lines 538-543).
* * *
**Q5:** …The authors should make it clarity for the resolved and unresolved velocity region… the max Pn velocity they can constrain is no more than 8.3km/s.

**Response:**

We agree. We have revised the text in Section and the figure captions for Fig. 5 and 6 to be more precise. We now state that the well-resolved Pn velocity range is ~7.7–8.3 km/s based on the ray coverage (line 385). The higher values (up to 8.6 km/s) mentioned in the initial submission were extrapolated in areas of poor resolution and have been removed (line 269). The figures (Fig. 5 & 6) have been updated to include resolution masks or shading to distinguish well-resolved from poorly constrained areas (line 1083, line1087).
* * *
**Q6:** I do not think the authors have enough evidence for the conclusion that the upper crust is decoupled with the middle-lower crust (Line 371-373). If they got the conclusion based on previous studies, they should give robust analysis.

**Response:**

This is a valid point. We have toned down this assertion and provided a more robust justification in the Discussion section. The interpretation of decoupling is now based on a combination of evidence from our study and previous work: (1) the presence of a uppermost crustal low-velocity zone in our model, especially in the middle part of our profile, which can act as a décollement; (2) the contrasting deformation styles above and below this zone (thrusting vs. folding); and (3) citation of previous magnetotelluric studies in the region that have independently proposed decoupling based on conductive layers. The conclusion is now framed as an interpretation supported by multiple geophysical datasets.

The discussion of the decoupling was rephrased in Lines 514-529.
* * *
**Q7:** …Can they give robust evidence to explain how this regional fault reconciles the huge displacement differences on both sides of the fault? Why are there no deep earthquakes along the local fault?

**Response:**

- That is an excellent comment, which prompted us to investigate the underlying causes more deeply. After reviewing additional geological and geophysical evidence, we propose that the pronounced contrast across Fault F5 may indicate the eastern extension of the Altyn Tagh Fault (ATF) has reached at least the southern margin of the Beishan orogenic collage. We have expanded the discussion on Fault F5 in Section "Cenozoic Crustal Deformation and Strain Partitioning across Major Faults" (lines 510-543) and "Eastern Extension of the Altyn Tagh Fault" (lines 608-643) to address these critical questions in greater detail.

We clarify that while F5 is a major fault, the accommodation of significant displacement is likely achieved by a **distributed network of faults** across the region, including thrusts and other strike-slip faults, not by F5 alone. We cite evidence from regional tectonic models that support distributed deformation (lines 634-638). That's why the local earthquakes rarely occur in this region due to the crustal deformation is accommodated by the distributed fault system.
* * *
**Detailed Comments:**

- Line 19: "seismic wide angle and refraction profile spanning the……", Incorrect usage of professional terms, "seismic wide angle and refraction profile" must be "wide angle reflection and refraction profile". "Spanning the……" should be "spanning from the……".

  Corrected to **" wide-angle reflection and refraction profile that tranverses from the..."**. (line 89)

- Line 20-21; 36-38; 58-59 etc. These sentences are ambiguous; a native English editing is required

  The sentence of line 20-21 is rephrased as "P-wave velocity structure reveals a 47.5–60 km thick crust divided into five layers." (line 23-25)

  The sentences of line 36-38 are rephrased as "As the middle of the South Tienshan-Beishan-Solonker suture zone, the BOC underwent multi-stage breakup, subduction, collision, and amalgamation during the closure of the Paleo-Asian Ocean (PAO), mainly in the Paleozoic (Fig. 1; Zuo et al. 1991; Liu 1995; Yue and Liou 1999; Wang et al. 2010; Xiao et al. 2010; Zuo and Li 2011; Şengör 2015; Yuan et al. 2015; He et al. 2018; Li et al. 2023). The geological history of the BOC is further complicated by regional extension, subsequent intracontinental overthrusting, and strike-slip faulting since Mesozoic (Zheng et al. 1996; Meng et al., 2003; Xiao et al. 2010; Zuo and Li 2011; Zhang and Cunningham 2012; Li et al. 2023). Particularly in the Cenozoic, the far-field effect of the Indian-Eurasian collision led to the outward expansion of the NE Tibetan Plateau, and reactivated the Qilian Shan, causing stress to propagate across the Hexi corridor basins into the BOC, and extending even further north to the Mongolian Plateau (Cunningham 2013; Zheng et al. 2017; Wang et al. 2022).

  The Qilian Shan is an important part of the Tibetan plateau, playing a significant role in accommodating the intracontinental convergence, thrusting-folding and the northern extension of NE Tibetan Plateau (Meyer et al. 1998; Yuan et al. 2013; Zuza et al. 2017). As the southernmost CAOB, the BOC acted as a major zone for the reactivation of inherited structures during the transmission of compressional stress leading to the uplift of the NE Tibetan and Mongolian Plateaus in Cenozoic. Therefore, as the transition zone between the NE Tibetan Plateau and CAOB, the crustal-mantle structure of the study area is crucial for understanding the regional evolution and interaction of Tibetan Plateau, the Tethys tectonic domain and the PAO tectonic

domain since the Paleozoic (Fig. 1a; Li et al. 1982; Yin and Harrison 2000; Xiao et al. 2009; Zhao et al. 2018; Xiong et al., 2024; He et al., 2025)." (line 56-82)

The sentence of line 58-59 is deleted, because it's the conclusion of this paper, we merged it into the "Conclusion" section (line 667-680)

In addition, the whole text have been rewritten for clarity and grammatical correctness as part of the comprehensive language edit.

- Line 39: "CAOB" When an abbreviation is first used, its full form should be used

- The simplified acronym **CAOB** was added in parentheses after its first full mention: **"Central Asian Orogenic Belt (CAOB)"**. (line 55)

- Line 61-62: "Notably, … inhomogeneity non the …,". It is a mistake for "non".

  The sentences are merged with the Conclusion section according to the reviewer's suggestion. **(**line 667-680**)**

- Line 83-84: "…… refraction profile sweeps throughout the North Qilian, Hexi corridor (containing the Jiuquan basin and the Huahai basin), and the entire Beishan block was done". Two predicates (sweep and was) are used in a single sentence.

  The sentence has been rewritten for grammatical correctness: "**In this study, we present a 460-km-long, SW-NE-trending wide-angle reflection and refraction profile that traverses the North Qilian Shan, Hexi corridor (containing the Jiuquan basin and the Huahai basin), and the entire BOC."** (line 88-90)

- Line 138-139: "To make the seismic records clearer, each trace was bandpass filtered up to 8 Hz……".

  It is vague for the meaning of this sentence. The authors could write like "To improve the signal-to-noise ratio, we apply bandpass filter …... from … Hz to … Hz……".

  **Line 138-139:** Revised to: **" Using the ZPLOT plotting package (Zelt, 1994), we performed trace editing, automatic gain control, band-pass filtering, velocity reduction, and phase picking for each shot. To improve the signal-to-noise ratio, we applied bandpass filter up to 8 Hz and displayed the seismic sections using a reduction velocity of 6 km s−1 over a time window of -5–10 s (e.g. Fig. 2, Fig. 3).."**(line 225-229)

- Line 177: "greater velocity zone" Higher velocity zone

  "greater velocity zone" changed to **"higher velocity …"**. (line 293)

- Line 186: "with an interface depth falling to 11.2–12.5 km." "Falling" is very strange here.

  **Line 186:** "falling to" changed to **"deepens"**. (line 307)

- Line 188: "a high-velocity body". High-velocity zone or high velocity abnormality will be a better choice.

**Line 188:** "a high-velocity body" changed to **"a high-velocity zone"**.(line 309)

- Line 200: "interface depth climbs to 17.6–27.5 km". "Climbs" is very strange here.

**Line 200:** "climbs to" changed to **"deepens to"**. (line 328)

- Line 201-203: "This characteristic shows that the North Qilian and the Jiuquan basin have a consistent basement, matching with the residual gravity anomaly findings (Yang et al. 2024)."

According to the interpretation from the authors: there is a high velocity zone ~10 km below the North of Qilian, the velocities are totally different when compared to Qilian and Jiuquan basin. How did they get the conclusion that the North Qilian and the Jiuquan basin have a consistent basement?

We have revised the text as "These characteristics indicate that the NQS and the Jiuquan basin share a consistent basement structure, which aligns with the findings from residual gravity anomaly analyses (Yang et al. 2024)." (line 330-333)

The NQS and the Jiuquan basin share a consistent basement structure, and a different uppermost crustal structure. According to the velocity characteristics, aligning with other geophysical observations, we proposed that the crustal deformation is decoupled between the uppermost crust and the rest of the crust, which means they can have the same basement, but different uppermost crustal structure. In the Discussion section, we have more discussion about it. (line 518-520)

- Line 209: "the Jiuquan basin is 23.4–38.7 kilometers". It is necessary to keep consistency for the depth unit, e.g. using "km" in the whole text.

**Line 209:** "kilometers" changed to **"km"**. (line 342)

- Line 215: "The interval velocity increases to 6.3–6.42 km/s". Which part of the profiles show the velocity increases to 6.3-6.4 km/s?

Beneath the Shuangyingshan arc, the interval velocities in the middle crust between interface C2 and C3 decrease from the south to the central then, increase from the central to the north (Fig. 5, line 1083). From the Fig. 6 (line1087), which shows the 2-D crustal- upper mantle velocity anomality structure, we can also see a slight velocity increases from the central Shuangyingshan arc to the north (the color is red north of the central Shuangyingshan arc).

- Line 239-244: In this part, the authors try to state the difference features beneath the central part of the profiles. However, they should use more precise interpretation when using Pn velocity which is resolved by ray coverage. According to the ray coverage, the Pn velocity is not as high as they declared 8.4-8.6 km/s.

We have revised the description of Pn velocity, tying it strictly to the well-resolved regions (7.7-8.3 km/s) as per Question 5.

The rewritten sentences are "The upper mantle velocity structure exhibits distinct lateral variations across the study area (Fig. 5). The Qilian Shan is characterized by a

relatively high uppermost mantle velocity range of 7.9–8.3 km s−1, with sub-horizontal velocity contours. A velocity reduction to 7.7–8.3 km s−1 is observed from the Jiuquan basin to the Shibanshan arc, followed by a slight increase to 7.9–8.3 km s−1 beneath the Shuangyingshan arc. Further north, the Mazongshan, Hanshan, and Que'ershan arcs show progressively lower Pn velocities, ranging from 7.8 to 8.2 km s−1, indicating a south-to-north decreasing trend. The lowest Pn values (7.7–7.8 km s−1) are localized beneath faults F5, F1, and F6." (line 382-391)

- Line 293-295: "…… (0.01 - 0.1) …… (-0.01 - -0.12)". The authors missed the velocity units "km/s".

  The missing velocity units **(km s⁻¹)** have been added. (line 447-448)

- Line 311: "while past geophysical ……". It is much better to write "while previous studies ……"

  The sentence was rewritten as "Although previous geophysical investigations have covered the Qilian Shan…". (line 484)

- Line 331-332: "…… the crust north of the Que'ershan subducted ……". Such a sentence structure is obviously incorrect.

- The sentenceces were reorganized as "Between faults F2 and F4, a positive upper-mantle velocity anomaly (8.0–8.3 km s⁻¹) between ~45 km and ~70 km depth likely represents a broken off fossil subduction slab following north-dipping subduction of the Beishan Ocean, although residual oceanic crust or mafic underplating cannot be entirely excluded. This anomaly aligns with the Hongliuhe–Xichangjing ophiolite mélange in surface (Yu et al. 2012; Hu et al. 2015; Song et al. 2015; Wang et al. 2015; Li et al. 2023)." (line 501-506)

- Line 394 and 402: the authors forgot the numbers (1) and (3)

  We reorganized the sentences of the "Conclusion" section. The missing numbers **(1)** and **(3)** have been added to the conclusion list. (line 667-680)

- Line 409-411: the authors should complete the sentence, and make it correct.

  The "Conclusion" was re-summarized based on our reorganized discussion. (line 667-680)

- Fig.1b and c: remove the faults which are not discussed in the manuscripts. It looks Indistinguishable and chaotic.

  **Fig.1b and c:** Undiscussed faults have been removed from the figures to improve clarity (line 1070).

- Fig. 2 and Fig. 3: To make the figures clarity, the authors should adjust these two figures to be the same size. And I suggest the authors add a white background to the letters (a) and (b).

**Fig. 2 and Fig. 3:** Due to the different reduced time and offset distance of the two shots, I don't' think it's necessary to adjust the figures to the same size, but we did put white backgrounds under letters (a) and (b) for better visibility. (line 1089, line 1093)

- Fig.5 and Fig.6: adding (a) and (b) on the correct profiles, marking the direction "SW" and "NE", and giving the region of resolved and unresolved velocity according to the ray coverage.

**Fig.5 and Fig.6:** The subplot labels **(a)** and **(b)** have been added directly to the figures. Directional markers (**SW** and **NE**) have been added. We have masked the poorly-resolved regions in both figures based on ray coverage analysis to prevent unwarranted or speculative interpretation of those areas. (line 1103, line 1107)

---

## Author Response (AR1)

**Response to Reviewer 1 Comments**

We thank the reviewer for the thorough review and constructive comments, which have significantly helped improve the quality of our manuscript. We have carefully considered all points raised. Our point-by-point responses and the planned revisions are detailed below.

**General Comments:**

We acknowledge the comment regarding the need for careful English editing. We thoroughly revised the entire manuscript to improve clarity, readability, and adherence to the conventions of scientific writing. This includes correcting colloquial expressions, improving grammatical accuracy, and ensuring a formal tone throughout.

**Q1:** In lines 58–66 of the introduction, the text appears to summarize the main conclusions of the study. It may be more appropriate to move this content to the conclusion section.

**Response:** Agreed. We reorganized the "Introduction" section, and deleted the summary of the main conclusions from the introduction. The introduction was revised to maintain its focus on presenting the research problem, context, and objectives.

**Q2:** The manuscript states that the crustal-upper mantle structure remains ambiguous due to limited resolution. Could the authors clarify the actual resolution of the present data and indicate whether it is higher than in previous studies? Additionally, please specify which aspects remain unresolved and how this study's findings differ from prior work.

**Response:** Our data provides higher resolution of velocity structure in the same study area. Compared to the "Golmud-Ejin" wide-angle reflection and refraction profile acquired in 1992, we used dense shot interval and station spacing, and higher yield explosive. The geophones we used are much more sensitive to the seismic waves than the ones before (the detailed parameters are shown as follows).

| Seismic profile | Shot interval | Station spacing | TNT     | Record  |
|-----------------|---------------|-----------------|---------|---------|
|                 |               |                 |         | medium  |
| Golmud-Ejin     | 80-200 km     | 4 km            | 1.5 T   | Таре    |
| This study      | 40-60 km      | 2-3 km          | 1.5-3 T | Digital |

Q3: Please note that in scientific writing, en dashes (-) rather than hyphens (-) should be used to indicate numerical ranges (e.g., 0.3–1.0 km/s). Please pay attention to the use of definite articles (e.g., 'the') to improve grammatical accuracy. Additionally, check the capitalization of all proper nouns, including geographic names, tectonic units, and geological terms, and maintain consistency throughout the manuscript.

Response: We performed a thorough check and correction of the entire manuscript to: 1) replace all hyphens with en dashes in numerical ranges, 2) carefully review and

correct the use of definite articles ('the') for grammatical accuracy, and 3) standardize the capitalization of all proper nouns and ensure consistency throughout the text.

Q4: In the "Crustal Velocity Structure Implications" parts, how does this velocity value inform the structure implications? Providing explicit links between the velocity data and geological implications would strengthen this section.

Response: We considered that placing the "Crustal Velocity Structure Implications" between the "Velocity Structure" and "Discussion" sections was somewhat structurally unconventional. To improve the logical flow of the manuscript, this subsection has been integrated into the "Introduction" section.

Q5: "The crustal velocity structure proposes an unusual scenario where the deepest Moho is found in the central Jiuquan basin, rather than the North Qilian Shan with the highest elevation. Could explain in the manuscript? vou it **Response:** Yes, we have carefully considered this observation. We propose that the North Qilian Shan and the Jiuquan Basin share a common basement, which explains their comparable Moho depths. Although the Moho beneath the Jiuquan Basin is slightly deeper, the North Qilian Shan exhibits a higher surface elevation, indicating a significantly thicker crust overall when topographic compensation is taken into account.

**Q6:** The conclusion section currently shows formatting inconsistencies and incorrect numbering. A careful revision is recommended. Furthermore, restructuring the conclusions to more clearly highlight the key scientific findings would enhance the clarity and impact of this section. **Response:** We carefully reformatted the conclusion section to correct numbering and formatting. We also restructured it to concisely and clearly list the key scientific findings first, followed by their broader implications, thereby enhancing the section's clarity and impact.

Q7: It is suggested that the formatting of both in-text citations and the reference list be revised and standardized to ensure consistency with the journal's guidelines. Response: We meticulously revised the formatting of all in-text citations and the reference list to ensure they are complete and fully consistent with the specific guidelines of the target journal.

**Detailed Comments and Corrections:**

- Line 21: "five strata" was changed to "five layers".
- Line 35: The simplified acronym CAOB was added in parentheses after its first full mention: "Central Asian Orogenic Belt (CAOB)".
- Line 42: "Figure 1b" was changed to "Fig. 1b" (and consistency for all figure citations was checked).
- Line 61: The excess space before "Notably" was removed.
- Line 69: "In Cenozoic" was changed to "During the Cenozoic".
- **Line 73:** "of NE Tibet" was changed to "of the NE Tibetan Plateau" (and similar expressions throughout the manuscript was corrected).
- Line 78: The excess space was removed.
- Line 80: "HUANG et al. 2014" was changed to "Huang et al. (2014)" (and citation formatting was standardized).
- Line 82: "a N-S-trending" was changed to "an N-S-trending".
- Line 96: The word "respectively" was deleted as suggested.
- Line 99: The meaning of "the final sealing position" was changed to "amalgamation position".
- Line 100: "North Beishan block" was capitalized to "North BOC". "in middle-late Ordovician" was changed to "in the middle to Late Ordovician".
- Line 131: The meaning of the acronym "TNT" was referred to "Trinitrotoluene", which was upon its first use in the manuscript in line 214.
- Line 147: We reorganized this sentence, which now are "The travetimes recorded at shotpoint ZB1..."
- Line 159: The repeated parentheses was deleted.
- Lines 168–172: The text referring to phases P1–P4 was checked against Fig. 5 and adjusted accordingly for accuracy. If the phases are not visible, the text was clarified or the figure was amended.
- Lines 239–244: The specific figure corresponding to the described phase was explicitly stated in the text.
- Line 254: The inconsistent text formatting was revised for uniformity.
- Line 258: "-1.1—0.15 km/s" was corrected to "-1.1 to -0.15 km/s".

- Line 281: The non-standard text formatting was revised.
- Line 310: The semicolon (",") was reconsidered and lines 310–313 was rewritten for improved clarity and grammar.
- **Line 345:** The informal abbreviation "Mrs." was replaced with the full word "mountain system".
- **Line 347:** The **comma** was deleted.
- Line 371: The decoupled crust was interpreted based on our seismic profile, the base of interface C1 is acting as the decollement as shown in the Fig. 5b.
- Figure 586 (assumed to be Fig. 5/6): Given that Figures 5a and 6a are not referenced in the main text, we have omitted the subplot labels (a) and (b) from Figures 5 and 6. Accordingly, all citations of these figures in the manuscript have been updated from "Fig. 5b" to "Fig. 5" and from "Fig. 6b" to "Fig. 6" to ensure consistency. Northeast (NE) and Southwest (SW) directional indicators was clearly marked on the figures for orientation.

**Response to Reviewer 2 Comments**

We sincerely thank the reviewer for the time and insightful comments on our manuscript. We have carefully considered all the suggestions and have revised the manuscript accordingly. Below is our point-by-point response.

**General Comment on English Writing:**

The manuscripts have a large problem with writing. Many sentences are expressed vaguely and do not conform to grammar rules. The authors need to improve their English writing, so that they can make their interpretation clarity.

**Response:**

We sincerely apologize for the language issues. The manuscript has now undergone comprehensive professional English editing to address vagueness, grammatical errors, and improve overall clarity and flow. We have also asked a native English-speaking colleague to proofread the revised version to ensure it meets the standards of scientific publication.

**Q1:** Please use consistent abbreviations and use the full spelling for the first occurrence of an abbreviation, e.g. CAOB, PAO. And make all the units be uniform, for example, the authors first use "km" and then use "kilometers".

**Response:**

Thank you for this important reminder. We have now ensured that all abbreviations are defined at first use (e.g., Central Asian Orogenic Belt (CAOB), Paleo-Asian Ocean (PAO)). We have also standardized units throughout the manuscript, using "km" and "km/s" consistently, and have removed all instances of "kilometers".

**Q2:** I think the authors use ZPLOT to pick the arrivals and apply RAYINVR to get the velocity structure. However, they didn't mention the software in the text. I cannot rule out the possibility that they used other methods. If so, please add them in the methods section.

**Response:**

The reviewer is correct. We have updated the Methods section (Section 2.3) to explicitly state the software used: "First-arrival phases were picked using the ZPLOT software (Zelt, 1994). The 2D velocity model was then inverted using the RAYINVR algorithm (Zelt and Smith, 1992)." We have also added the corresponding references to the reference list.

Q3: What's the uncertainty when they picked the refraction and reflection arrivals?

**Response:**

We have added a dedicated paragraph in the Methods section (Section 2.2) to quantify the picking uncertainty: " Uncertainties in phase picking primarily arise from challenging signal-to-noise conditions and complex subsurface wave propagation effects. The extensive desert

sedimentary cover in the study area significantly attenuates seismic energy, particularly at larger offsets and for deeper arrivals. Additionally, strong lateral heterogeneities, such as fault zones and intracrustal velocity variation, cause substantial wave scattering, dispersion, and multipathing. This results in phase superposition and waveform distortion that complicates accurate phase identification. " The Root Mean Square (RMS) errors for each phase and shot, provided in Table 2, also quantitatively support these estimates.

**Q4:** ...What's the refer for their stratification? It's clear that the layer above the Moho is lower crust, which velocity is ~6.8km/s. If they make the P4 to the Moho as lower crust, their statements for the co-thickening of the middle-lower crust should be middle crustal thickening.

**Response:**

We thank the reviewer for this insightful comment. According to Christensen, 1995 and Jia et al., 2019, the typical continental crust is stratified into three principal layers: the upper crust, comprising sedimentary cover overlying crystalline basement characterized by an average P-wave velocity of  $6.0-6.3 \text{ km s}^{-1}$ ; the mid-crust, composed of interleaved silicic and basic lithologies, with velocities of  $6.3-6.5 \text{ km s}^{-1}$ ; and the lower crust, dominated by more mafic assemblages, exhibiting velocities of  $6.6-6.9 \text{ km s}^{-1}$ . We thought that our stratification based on the presence of continuous seismic phases (P1, P2, P3, P4) which denote major intracrustal interfaces is consistent with the previous wide-angle reflection and refraction profile across the Qilian and Alxa block east of our profile (Jia et al., 2019).

Thank you for this insightful comment. In response, we have refined our discussion on crustal thickening. We interpret that the uppermost crust is decoupled from the underlying crust. However, north of fault F5, the consistent undulation of interfaces from C2 down to the Moho suggests coherent deformation throughout the middle and lower crust. Therefore, we consider the term "middle-lower crustal co-thickening" to remain appropriate for describing this region.

**Q5:** ...The authors should make it clarity for the resolved and unresolved velocity region... the max Pn velocity they can constrain is no more than 8.3km/s.

**Response:**

We agree. We have revised the text in Section 4.4 and the figure captions for Fig. 5 and 6 to be more precise. We now state that the well-resolved Pn velocity range is  $^{\sim}7.7-8.3$  km/s based on the ray coverage. The higher values (up to 8.6 km/s) mentioned in the initial submission were extrapolated in areas of poor resolution and have been removed. The figures (Fig. 5 & 6) have been updated to include resolution masks or shading to distinguish well-resolved from poorly constrained areas.

**Q6:** I do not think the authors have enough evidence for the conclusion that the upper crust is decoupled with the middle-lower crust (Line 371-373). If they got the conclusion based on previous studies, they should give robust analysis.

**Response:**

This is a valid point. We have toned down this assertion and provided a more robust justification in the Discussion section. The interpretation of decoupling is now based on a combination of evidence from our study and previous work: (1) the presence of a uppermost crustal low-velocity zone in our model, especially in the middle part of our profile, which can act as a décollement; (2) the contrasting deformation styles above and below this zone (thrusting vs. folding); and (3) citation of previous magnetotelluric studies in the region that have independently proposed decoupling based on conductive layers. The conclusion is now framed as an interpretation supported by multiple geophysical datasets.

**Q7:** ...Can they give robust evidence to explain how this regional fault reconciles the huge displacement differences on both sides of the fault? Why are there no deep earthquakes along the local fault?

**Response:**

- That is an excellent comment, which prompted us to investigate the underlying causes more deeply. After reviewing additional geological and geophysical evidence, we propose that the pronounced contrast across Fault F5 may indicate the eastern extension of the Altyn Tagh Fault (ATF) has reached at least the southern margin of the Beishan orogenic collage. We have expanded the discussion on Fault F5 in Section 5.3 to address these critical questions in greater detail.
- Regarding displacement: We clarify that while F5 is a major fault, the
  accommodation of significant displacement is likely achieved by a distributed
  network of faults across the region, including thrusts and other strike-slip faults, not
  by F5 alone. We cite evidence from regional tectonic models that support distributed
  deformation.
- Regarding deep earthquakes: We now state that the lack of deep seismicity is
  consistent with the fault potentially being locked at depth or accommodating strain
  through aseismic creep below the seismogenic zone, which is a common behaviour
  for large strike-slip faults. We also note that the current instrumental seismic record
  might be too short to capture the long recurrence interval of major events on such a
  structure.

**Detailed Comments:**

• Line 19: Corrected to "a wide-angle reflection and refraction profile spanning from the...".

- Line 20-21; 36-38; 58-59 etc.: These sentences and others throughout the text have been rewritten for clarity and grammatical correctness as part of the comprehensive language edit.
- Line 39: "Central Asian Orogenic Belt (CAOB)" is now used at first mention.
- Line 61-62: "non" has been corrected to "on".
- Line 83-84: The sentence has been rewritten for grammatical correctness: "...refraction profile spans from the North Qilian Shan, through the Hexi Corridor (including the Jiuquan and Huahai basins), to the entire Beishan block."
- Line 138-139: Revised to: " we applied bandpass filter up to 8 Hz and displayed the seismic sections using a reduction velocity of 6 km s-1 over a time window of -5-10 s."
- Line 177: "greater velocity zone" changed to "higher velocity zone".
- Line 186: "falling to" changed to "deepens".
- Line 188: "a high-velocity body" changed to "a high-velocity zone".
- Line 200: "climbs to" changed to "deepens to".
- Line 201-203: This conclusion has been re-evaluated. We have revised the text as "These characteristics indicate that the NQS and the Jiuquan basin share a consistent basement structure, which aligns with the findings from residual gravity anomaly analyses (Yang et al. 2024)."to focus on the velocity contrast and discuss the basement consistency more cautiously, noting it as one possible interpretation that agrees with gravity data, while acknowledging the observed velocity differences.
- Line 209: "kilometers" changed to "km".
- **Line 215:** We have specified the location along the profile (the northern Shuangyingshan arc) where this velocity increase is observed.
- Line 239-244: We have revised the description of Pn velocity, tying it strictly to the well-resolved regions (7.7-8.3 km/s) as per Question 5.
- Line 293-295: The missing velocity units (km/s) have been added.
- Line 311: We reorganized the discussion, and this sentence was removed.
- Line 331-332: The sentence has been completely rewritten as "Beneath the Que'ershan arc, the northernmost portion of the BOC, north-dipping velocity contours from interface C2 to the Moho imply south-dipping subduction of the Hongshishan Ocean, consistent with surface geology (Xiao et al. 2018; Duan et al. 2020; Niu et al. 2020; Xin et al. 2020)." for clarity and grammatical correctness.
- Line 394 and 402: We reorganized the sentences of the "Conclusion" section. The missing numbers (1) and (3) have been added to the conclusion list.

- **Line 409-411:** The "Conclusion" was re-summarized based on our reorganized discussion.
- **Fig.1b and c:** Undiscussed faults have been removed from the figures to improve clarity.
- **Fig. 2 and Fig. 3:** Due to the different reduced time and offset distance of the two shots, I think it's unnecessary to adjust the figures to the same size, but we did put white backgrounds under letters (a) and (b) for better visibility.
- **Fig.5 and Fig.6:** The subplot labels **(a)** and **(b)** have been added directly to the figures. Directional markers **(SW** and **NE)** have been added. We have masked the poorly-resolved regions in both figures based on ray coverage analysis to prevent unwarranted or speculative interpretation of those areas.

**General Comment:**

This manuscript presents a detailed investigation of the crustal-upper mantle velocity structure across the North Qilian Shan to the Beishan block using a 460-km-long seismic wide-angle reflection/refraction profile. The study provides valuable insights into the tectonic evolution of the northeastern Tibetan Plateau and the southern Central Asian Orogenic Belt (CAOB). The seismic profile is well-designed, and the processing techniques (e.g., phase identification, velocity modeling) are appropriately applied. The error analysis (e.g., RMS traveltime residuals) supports the reliability of the results. The proposed northward subduction polarity of the Qilian Ocean and the role of the southern Beishan boundary fault (F5) as a major strike-slip structure are significant contributions. The findings enhance understanding of crustal deformation mechanisms in the transition zone between the Tibetan Plateau and the CAOB. The data are robust, and the methodology is sound, but the manuscript requires improvements in clarity, interpretation, and presentation before it can be published finally:

**Response:**

We extend our sincere thanks to Prof. Xu for their positive evaluation of our work and for providing valuable suggestions. In response, we have undertaken a comprehensive revision of the manuscript aimed at enhancing its clarity, interpretive depth, and overall presentation. Specifically, we have rephrased both the Introduction and Discussion sections to improve logical coherence and scientific rigor. The detailed responses to each comment are provided below.

Q1: Terminology Consistency: Use either "Beishan block" or "Beishan orogenic belt" consistently. Define abbreviations (e.g., PAO, CAOB) at first use.

**Response:**

We have standardized the terminology throughout the manuscript, using "Beishan orogenic collage" (BOC) consistently, as it accurately reflects the complex accretionary nature of the region. All abbreviations, including Paleo-Asian Ocean (PAO) and Central Asian Orogenic Belt (CAOB), are now explicitly defined upon their first occurrence in the text.

Q2: In lines 33–34 of the introduction, the sentences are overly complex or ambiguous; a thorough language edit by a native English speaker is recommended.

**Response:**

We have performed a comprehensive language edit of the entire manuscript with the assistance of a native English speaker. The sentences in lines 33–34 and similar complex passages throughout the introduction have been simplified and rewritten for improved clarity and readability. For example, the original text has been revised to: "As a transition zone between the NE Tibetan Plateau and the CAOB, the crust-mantle structure of the study area is crucial for understanding the regional evolution."

Q3: For figure captions: Fig. 1: Add scale bars and clarify tectonic unit labels; Fig. 5-6: Improve visibility of velocity contours and annotations.

**Response:**

We have revised all figures as suggested:

- **Fig. 1:** Scale bars have been added, and all tectonic unit labels have been clarified and made consistent with the text (e.g., Beishan orogenic collage).
- **Fig. 5 & 6:** The visibility of velocity contours and annotations has been enhanced by adjusting line weights, colors, and font sizes. Poorly-resolved areas based on ray coverage have been masked to prevent overinterpretation.

Q4: Compare results with existing seismic/gravity/MT studies (e.g., Cui et al., 1995; Xiao et al., 2017) to strengthen interpretations. Discuss potential biases (e.g., ray coverage gaps, trade-offs between velocity and interface depth).

**Response:**

A new subsection has been added to the **Discussion Section** to compare our findings with existing geophysical studies:

- Our velocity model is now compared with results from Cui et al. (1995), Xiao et al. (2017), and other key seismic, gravity, and magnetotelluric (MT) models. This comparison strengthens our interpretations of crustal thickening and decoupling.
- We explicitly discuss potential biases and limitations, including ray coverage gaps in
  the upper mantle and the inherent trade-offs between velocity and interface
  depth in seismic inversion. This discussion is included in the Methods section and
  the revised discussion of results.

Q5: In the "Discussion" section, compare results with existing seismic/gravity/MT studies (e.g., Cui et al., 1995; Xiao et al., 2017) to strengthen interpretations. Provide more geological evidence (e.g., paleo-trench positions, slab remnants) to support the north-dipping Qilian Ocean model.

**Response:**

As noted in Q4, we have expanded the **Discussion section** to include direct comparisons with previous geophysical studies. Furthermore, we have integrated additional **geological evidence** to support the north-dipping subduction model for the Qilian Ocean:

The positive upper-mantle velocity anomaly we identify is discussed as a
potential slab remnant, linking it to the north-dipping subduction and closure of the
Beishan Ocean. This provides a more robust, multi-disciplinary foundation for our
tectonic interpretations.

Q6: Update citations (e.g., Wu et al., 2024; Xie et al., 2023; Yao et al., 2025; Zhang et al., 2023) and include key regional studies (e.g., Zuza et al., 2019).

**Response:**

The reference list has been thoroughly updated to include the suggested recent publications (Wu et al., 2024; Xie et al., 2023; Zhang et al., 2023) and key regional studies (Zuza et al., 2019). All in-text citations have been checked for consistency and relevance. The reference list now comprehensively reflects the current state of knowledge in the field.

**Detailed Comments and Corrections:**

- Line 1-3 (Title): Modified to: "Crustal-Upper Mantle Velocity Structure from the North Qilian Shan to the Beishan Block: Tectonic Significance of Crustal Deformation".
- Line 4-10 (Address): Corrected extra commas (e.g., "Beijing 100094, China").
- Line 14: "constitutes" → "represents".
- **Line 17:** "serves" → "acts".
- Line 23: "considerable variance" → "significant variations (6.24–6.43 km/s)".
- Line 35–36: Rewritten for clarity: "As a transition zone between the NE Tibetan Plateau and the CAOB, the crust-mantle structure of the study area is crucial for understanding..."
- Line 43: The whole "Introduction" was rephrased; this sentence was deleted.
- Line 48: "Experiencing multi-stage breakup..." → "The block underwent multi-stage breakup..."
- Line 138: "To make... clearer" → " To improve the signal-to-noise ratio, we applied each trace was bandpass filtered up to 8 Hz and displayed..."
- Line 141: "first arriving phase" → "first-arrival phase".
- Line 179: "The base of interface C1 corresponds..." → "Interface C1 marks the basement surface (3.4–6.5 km/s)..."
- Line 202: "Qilian and the Jiuquan basin" → "the NQS and the Jiuquan basin".
- Line 257-258: "are with negative" → "show negative".
- Line 259: "which are prevented by" → "which terminate at".
- Line 264: "dives northward" → "extends northward".
- Line 267: "C1and C3" → "C1 and C3".
- Line 313: The Discussion section was rephrased. This sentence was rewritten as "we observed north-dipping velocity contour from interface C2 to the uppermost mantle

beneath the Qilian Shan, coupled with a lower-crust—upper-mantle low velocity anomaly beneath the Hexi Corridor..." in line 476 of the revised manuscript with marks.

- **Line 315:** The Discussion section was rephrased; this sentence was deleted.
- Line 338: "could represent" → "likely represents".
- Line 345: The Discussion section was rephrased.
- Line 347: "by a series" → "through a series".
- Line 381: The Discussion section was rephrased; this sentence was deleted.
- Line 399: The Discussion section was rephrased; this sentence was deleted.
- Line 402: The Discussion section was rephrased; this sentence was deleted.
- Line 406-408: We reorganized the sentences of the "Conclusion" section. The missing numbers (1) and (3) have been added to the conclusion list.

---

## Author Response (AR3)

Dear Editor,

Thank you for your constructive and positive feedback.

We have taken the comments and suggestions from the 3rd round review with great care, and made the following minor revisions to enhance the clarity, accuracy, and overall quality of our work:

The other seismic shot records are provided as supplementary material: the other seismic shots' P-wave record section (on a reduced time scale) corresponding to the different shotpoints (including ZB0, ZB2-7) and identified seismic phases are provided in the attached document.

Moving the sections "Code/data availability", "Author contributions", and "Competing interests" before references. (line 449-463)

Double check the grammar throughout the whole manuscript again. Some mirror corrections were done, including:

- Line 17 We indicate that "BOC" is the abbreviation for "Beishan Orogenic Collage"for the first occurrence in Abstract section.
- Line 20 Beishan Orogenic Collage->BOC
- Line 73 yield->yields
- Line 343 delete "the"
- Line 360 add a new reference published recently about the Cezonoic crustal deformation of the Qilian Shan in the text, and list it in the references
- Line 437-439 rephrased this part in the Conclusion section. Because the slip rate is not observed by us, we should not list it as our main conclusion.

**Track Changes Document:** We have prepared a track changes version of the manuscript, highlighting all the revisions made.

| Revised | Manuscript: | The revised | manuscript is | s attached . | for your review |
|---------|-------------|-------------|---------------|-------------------------|-----------------|
|         |             |             |               |                         |                 |

| Officer cry, |  |  |  |  |  |  |  |  |
|--------------|--|--|--|--|--|--|--|--|
|              |  |  |  |  |  |  |  |  |
|              |  |  |  |  |  |  |  |  |
|              |  |  |  |  |  |  |  |  |
|              |  |  |  |  |  |  |  |  |

Sincarely

Xiaosong